# Multilayer Matrix Factorization via Dimension-Reducing Diffusion Variational Inference

Junbin Liu [1]   Farzan Farnia [2]   Wing-Kin Ma [1]

## Abstract

Multilayer matrix factorization (MMF) has recently emerged as a generalized model of, and potentially a more expressive approach than, the classic matrix factorization. This paper considers MMF under a probabilistic formulation, and our focus is on inference methods under variational inference. The challenge in this context lies in determining a variational process that leads to a computationally efficient and accurate approximation of the maximum likelihood inference. One well-known example is the variational autoencoder (VAE), which uses neural networks for the variational process. In this work, we take insight from variational diffusion models in the context of generative models to develop variational inference for MMF. We propose a dimension-reducing diffusion process that results in a new way to interact with the layered structures of the MMF model. Experimental results demonstrate that the proposed diffusion variational inference method leads to improved performance scores compared to several existing methods, including the VAE.

## 1. Introduction

Over decades, matrix factorization (MF) methods have played a crucial role in a wide variety of problems such as dimensionality reduction, low-dimension representation learning, blind source separation (Hyvärinen et al., 2023), hyperspectral unmixing (Ma et al., 2013), topic modeling (Arora et al., 2012), community detection (Yang & Leskovec, 2013), to name a few. The broad interest of researchers in this subject has led to both diverse and substantial developments. In particular, we have seen different ways to exploit the hidden structures of the underlying matrix factors, such as statistical independence, sparsity, and non-negativity. Some of such methods are equipped with desirable results such as identifiability guarantees—that is, the guarantees of identifying the underlying ground-truth factors—which are essential in applications such as blind source separation; see, e.g., (Gillis, 2020; Khemakhem et al., 2020; Wu et al., 2021) and the references therein. MF is intimately linked with the notion of learning low-dimensional structures from higher-dimensional data. It is closely related to latent-variable component analysis such as independent component analysis (ICA).

More recently, there has been interest in multilayer, and possibly nonlinear, MF (Trigeorgis et al., 2016; Zhao et al., 2017; Xue et al., 2017; Fan, 2021; De Handschutter & Gillis, 2023). Multilayer MF (MMF) is a more general model than the (two-factor) MF model, and it is anticipated that MMF should provide more powerful results. It was empirically shown that MMF can extract meaningful hierarchical features from data, which offers new insights in applications such as clustering; see, e.g., (Trigeorgis et al., 2016; De Handschutter et al., 2021) and the references therein. MMF is also related to nonlinear latent-variable component analysis (Khemakhem et al., 2020). In particular, if the nonlinear system is modeled by a neural network, we can see it as a multilayer system.

One powerful approach to MF or MMF is to formulate the factorization model as a latent-variable model and treat the factorization problem as a probabilistic inference problem. In this direction, variational inference (VI) has been found to be promising in providing a practical way to handle complex models (Rezende et al., 2014; Ranganath et al., 2015). In particular, for MMF, variational autoencoders (VAEs) appear to be the only available solution in the literature so far; see (Khemakhem et al., 2020) and also (Li et al., 2024).

Lately, diffusion models (DMs) (Sohl-Dickstein et al., 2015; Ho et al., 2020; Song et al., 2021; Luo, 2022) have caught tremendous attention in the context of generative models. They have been found to provide competitive performance in various generation tasks. There are several ways to derive

[1]Department of Electronic Engineering, the Chinese University of Hong Kong, Hong Kong SAR of China [2]Department of Computer Science and Engineering, the Chinese University of Hong Kong, Hong Kong SAR of China. Correspondence to: Junbin Liu <liujunbin@link.cuhk.edu.hk>, Farzan Farnia <farnia@cse.cuhk.edu.hk>, Wing-Kin Ma <wkma@ee.cuhk.edu.hk>.

*Proceedings of the 42nd International Conference on Machine Learning*, Vancouver, Canada. PMLR 267, 2025. Copyright 2025 by the author(s).

and understand DMs: We can consider stochastic differential equations, seeking to reverse a diffusion process (Song et al., 2021); DMs can also be seen as an outcome of denoising score matching and Langevin dynamics for learning the data distribution and generating data (Song & Ermon, 2019). DMs can also be derived by formulating a latent-variable model and then by performing a specific (diffusion) type of VI (Ho et al., 2020; Kingma et al., 2021)—we are most attracted by this interpretation. Given the success of DMs, we consider this question: Can we take the VI in DMs and apply it to MMF?

So far, and to our best knowledge, DM-based VI has not been considered for MMF. And DM-based VI cannot be directly applied to MMF. This is because the current DMs assume equal dimensions with the latent variables, while MMF has unequal latent-variable dimensions. In this paper, we explore the application of DM-based VI to MMF. We will propose a dimension-reducing (DR) variational diffusion model. The distinct characteristic is that we associate each layer of the DM with a layer of the MMF model, and we seek to use light-weight methods to deal with each layer. This is different from DMs and hierarchical VAEs (Ranganath et al., 2016; Sønderby et al., 2016; Vahdat & Kautz, 2020) for generative models, which would employ deep neural networks at each layer. From the proposed DR diffusion model, we will derive a VI scheme. Numerical results will be provided to demonstrate the performance of the proposed DR diffusion VI (DRD-VI).

It is worth noting that, in the context of generative models, there are studies that consider dimensionality reduction for diffusion models. In (Rombach et al., 2022; Wang et al., 2023), the authors apply dimensionality reduction before the diffusion model. In (Jing et al., 2022; Zhang et al., 2023), the authors concatenate multiple diffusion models, and at each stage dimensionality reduction is applied. In the aforementioned studies, dimensionality reduction is done outside of the diffusion process. Our study differs in that we embed dimensionality reduction inside the diffusion model.

In addition it is interesting to have a comparison with the hierarchical VAE (HVAE) approach (Sønderby et al., 2016), which, similar to the DM, is also capable of interacting with the layered structures of the model. The HVAE was not considered in the context of MMF, to the best of our knowledge, although in principle it is possible to do so. As noted earlier, the HVAE employs a deep network for each layer of the variation process. While this makes the variational process more powerful, it also makes the HVAE more difficult to train. Moreover, it was argued that the stochastic approximation in the HVAE may have larger variance as the number of layers is larger (Luo, 2022). In comparison, our proposed DRD-VI adopts light-weight operations at each layer of the variational process, which in turn makes the training easier. This is an advantage that has been noted in the context of generative models; see (Luo, 2022).

## 2. Background

### 2.1. Multilayer Matrix Factorization

Consider the following problem. Let $\boldsymbol{y} \in \mathbb{R}^M$ denote a data point. It is modeled to follow a generative model

$$\boldsymbol{y} = f_{\boldsymbol{\theta}}(\boldsymbol{z}) + \boldsymbol{v}, \tag{1}$$

where $f_{\boldsymbol{\theta}} : \mathbb{R}^N \to \mathbb{R}^M$ is a function parameterized by $\boldsymbol{\theta}$, $\boldsymbol{z} \in \mathbb{R}^N$ is the latent variable associated with $\boldsymbol{y}$, whose dimension $N$ is assumed to be less than the data dimension $M$; $\boldsymbol{v}$ is noise and is modeled as $\boldsymbol{v} \sim \mathcal{N}(\boldsymbol{0}, \sigma^2 \boldsymbol{I})$. Let $\{\boldsymbol{y}_1, \boldsymbol{y}_2, \ldots, \boldsymbol{y}_L\}$ be a given set of data points that follow the model in (1) and are independently distributed. Our goal is to estimate $\boldsymbol{\theta}$ from $\{\boldsymbol{y}_1, \ldots, \boldsymbol{y}_L\}$ and then to estimate the latent variable $\boldsymbol{z}_n$ of each $\boldsymbol{y}_n$.

If $f_{\boldsymbol{\theta}}$ takes a linear form $f_{\boldsymbol{\theta}}(\boldsymbol{z}) = \boldsymbol{A}\boldsymbol{z}$, with $\boldsymbol{\theta} = \{\boldsymbol{A}\}$, then the problem can be seen as a matrix factorization problem. In particular, by letting $\boldsymbol{Y} = [\boldsymbol{y}_1, \ldots, \boldsymbol{y}_L]$ and $\boldsymbol{z} = [\boldsymbol{z}_1, \ldots, \boldsymbol{z}_L]$, the problem is essentially to recover $\boldsymbol{A}$ and $\boldsymbol{Z}$ from $\boldsymbol{Y}$ such that $\boldsymbol{Y} \approx \boldsymbol{A}\boldsymbol{Z}$. In recent years, we have seen interest in multilayer matrix factorization (MMF), which considers

$$f_{\boldsymbol{\theta}}(\boldsymbol{z}) = \boldsymbol{A}_1 \boldsymbol{A}_2 \ldots \boldsymbol{A}_T \boldsymbol{z}, \quad \boldsymbol{\theta} = \{\boldsymbol{A}_1, \boldsymbol{A}_2, \ldots, \boldsymbol{A}_T\},$$

and leads to a multilayer factorization $\boldsymbol{Y} \approx \boldsymbol{A}_1 \ldots \boldsymbol{A}_T \boldsymbol{Z}$. In MMF, we would impose structures at each layer. Let $\boldsymbol{x}_t = \boldsymbol{A}_{t+1} \ldots \boldsymbol{A}_T \boldsymbol{z}$. In non-negative MMF, we constrain $\boldsymbol{x}_t \geq \boldsymbol{0}$ (as well as $\boldsymbol{z} \geq \boldsymbol{0}$) (Trigeorgis et al., 2016). Such MMF was numerically demonstrated to provide meaningful results in learning attribute representations of images. We have also seen interest in the following model:

$$f_{\boldsymbol{\theta}}(\boldsymbol{z}) = \rho(\boldsymbol{A}_1 \rho(\boldsymbol{A}_2 \rho(\ldots \rho(\boldsymbol{A}_T \boldsymbol{z}) \ldots))), \tag{2}$$

where $\rho$ is a component-wise nonlinear activation function. The function $\rho$ is used to impose structures at each layer; e.g., if $\rho$ is a ReLU function, we enforce non-negativity with each layer's output. The model in (2) can be viewed as a neural network, modeling a nonlinear relationship between $\boldsymbol{y}$ and $\boldsymbol{z}$. In this sense, we are also dealing with a nonlinear latent-variable component analysis problem; see, e.g., (Khemakhem et al., 2020). Additionally, it was argued that nonlinear factorization $f_{\boldsymbol{\theta}}(\boldsymbol{z}) = \rho(\boldsymbol{A}\boldsymbol{z})$ provides an effective model for low-dimensional embedding of high-dimensional data (Saul, 2022).

### 2.2. Variational Inference for MMF

We consider a probabilistic framework for MMF. Consider the generative model in (1) and (2). Assume that the latent variable $\boldsymbol{z}$ follows a known distribution $p(\boldsymbol{z})$, called

the latent prior. For example, in independent component analysis (ICA) the latent prior is chosen as a component-wise independent and non-Gaussian distribution. In simplex component analysis (SCA) (Wu et al., 2021), an important type of non-negative matrix factorization, the latent prior may be chosen as a simplex uniform distribution

$$p(\boldsymbol{z}) = \mathbb{1}_{\Delta}(\boldsymbol{z})/Z, \qquad (3)$$

where $\Delta = \{\boldsymbol{z} \in \mathbb{R}^N \mid \boldsymbol{z} \geq \boldsymbol{0}, \boldsymbol{1}^{\top}\boldsymbol{z} = 1\}$ is a unit simplex; $\mathbb{1}_{\mathcal{X}}$ is an indicator function ($\mathbb{1}_{\mathcal{X}}(\boldsymbol{x}) = 1$ if $\boldsymbol{x} \in \mathcal{X}$ and $\mathbb{1}_{\mathcal{X}}(\boldsymbol{x}) = 0$ if $\boldsymbol{x} \notin \mathcal{X}$); $Z$ is a normalizing constant. We can also consider a non-negative bounded uniform distribution

$$p(\boldsymbol{z}) = \mathbb{1}_{[0,1]^N}(\boldsymbol{z}) \qquad (4)$$

for non-negative matrix factorization. The distribution of the data point $\boldsymbol{y}$ can be expressed as

$$p_{\boldsymbol{\theta}}(\boldsymbol{y}) = \int p_{\boldsymbol{\theta}}(\boldsymbol{y}|\boldsymbol{z})p(\boldsymbol{z})d\boldsymbol{z} = \mathbb{E}_{p(\boldsymbol{z})}[p_{\boldsymbol{\theta}}(\boldsymbol{y}|\boldsymbol{z})], \quad (5)$$

where $p_{\boldsymbol{\theta}}(\boldsymbol{y}|\boldsymbol{z}) = \mathcal{N}(\boldsymbol{y}; f_{\boldsymbol{\theta}}(\boldsymbol{z}), \sigma^2\mathbf{I})$.

We want to estimate $\boldsymbol{\theta}$ from a given dataset $\{\boldsymbol{y}_1, \ldots, \boldsymbol{y}_L\}$. We pursue maximum-likelihood (ML) estimation. Let

$$\mathcal{L}(\boldsymbol{\theta}; \boldsymbol{y}) = \log p_{\boldsymbol{\theta}}(\boldsymbol{y}),$$

denote the log-likelihood function for $\boldsymbol{y}$. ML estimation determines $\boldsymbol{\theta}$ by solving

$$\max_{\boldsymbol{\theta}} \sum_{n=1}^{L} \mathcal{L}(\boldsymbol{\theta}; \boldsymbol{y}_n).$$

The challenge with the ML estimation problem above is that the log-likelihood $\mathcal{L}(\boldsymbol{\theta}; \boldsymbol{y})$ has no known tractable expression in general; this is because (5) is a multi-dimensional integral that has no closed-form or explicit equation in general. One can approximate (5) by a stochastic (Monte Carlo sampling) approximation method, but such methods were often found to be computationally inefficient in practice.

Recent research has considered variational inference (VI), together with stochastic approximation, as a more practical way to approximate the log-likelihood function. Let $q_{\boldsymbol{\phi}}(\boldsymbol{z}|\boldsymbol{y})$ be some distribution function with parameter $\boldsymbol{\phi}$, which will be called the variational distribution in the sequel. Consider the Jensen inequality

$$\mathcal{L}(\boldsymbol{\theta}; \boldsymbol{y}) \geq \widehat{\mathcal{L}}(\boldsymbol{\theta}, \boldsymbol{\phi}; \boldsymbol{y}) = \mathbb{E}_{q_{\boldsymbol{\phi}}(\boldsymbol{z}|\boldsymbol{y})}\left[\log \frac{p_{\boldsymbol{\theta}}(\boldsymbol{y}|\boldsymbol{z})p(\boldsymbol{z})}{q_{\boldsymbol{\phi}}(\boldsymbol{z}|\boldsymbol{y})}\right].$$

The function $\widehat{\mathcal{L}}$ is called the evidence lower bound (ELBO). The idea is to choose a $q_{\boldsymbol{\phi}}$ such that $\widehat{\mathcal{L}}$ would be computationally efficient to compute or approximate. We also

hope that the choice of $q_{\boldsymbol{\phi}}$ would lead to a small gap between $\mathcal{L}$ and $\widehat{\mathcal{L}}$, and thereby a good approximation of the log-likelihood function.

Take the famous variational autoencoder (VAE) as an example. The latent prior is Gaussian, specifically, $p(\boldsymbol{z}) = \mathcal{N}(\boldsymbol{z}; \boldsymbol{0}, \mathbf{I})$. The variational distribution is chosen as $q_{\boldsymbol{\phi}}(\boldsymbol{z}|\boldsymbol{y}) = \mathcal{N}(\boldsymbol{z}; \boldsymbol{\mu}_{\boldsymbol{\phi}}(\boldsymbol{y}), \text{Diag}(\boldsymbol{\sigma}_{\boldsymbol{\phi}}^2(\boldsymbol{y})))$, where $\boldsymbol{\mu}_{\boldsymbol{\phi}}$ and $\boldsymbol{\sigma}_{\boldsymbol{\phi}}$ are neural networks with parameter $\boldsymbol{\phi}$. By the parameterization trick (see (Kingma & Welling, 2013) for details), it was found that $\widehat{\mathcal{L}}$ can be efficiently handled by stochastic approximation. This VAE approach has been employed in ICA and SCA (Khemakhem et al., 2020; Li et al., 2024).

Let us write down the VI problem:

$$\max_{\boldsymbol{\theta}, \boldsymbol{\phi}} \frac{1}{L} \sum_{n=1}^{L} \widehat{\mathcal{L}}(\boldsymbol{\theta}, \boldsymbol{\phi}; \boldsymbol{y}_n).$$

Note that the variational model parameter $\boldsymbol{\phi}$ is also optimized for best ELBO approximation given the structure of $q_{\boldsymbol{\phi}}$. Additionally, we should mention the estimation of the latent variables once $(\boldsymbol{\theta}, \boldsymbol{\phi})$ is obtained from the VI problem. Consider the minimum mean square error (MMSE) estimate $\widehat{\boldsymbol{z}}_n = \mathbb{E}_{p_{\boldsymbol{\theta}}(\boldsymbol{z}|\boldsymbol{y}_n)}[\boldsymbol{z}]$. There is no known tractable equation for $p_{\boldsymbol{\theta}}(\boldsymbol{z}|\boldsymbol{y})$. The variational distribution $q_{\boldsymbol{\phi}}(\boldsymbol{z}|\boldsymbol{y})$ can be seen as an approximation of $p_{\boldsymbol{\theta}}(\boldsymbol{z}|\boldsymbol{y})$ because the ELBO attains equality if and only if $q_{\boldsymbol{\phi}}(\boldsymbol{z}|\boldsymbol{y}) = p_{\boldsymbol{\theta}}(\boldsymbol{z}|\boldsymbol{y})$. This leads us to employ

$$\widehat{\boldsymbol{z}}_n = \mathbb{E}_{q_{\boldsymbol{\phi}}(\boldsymbol{z}|\boldsymbol{y}_n)}[\boldsymbol{z}]. \qquad (6)$$

## 3. Dimension-Reducing Diffusion VI for MMF

Our endeavor is to take insight from variational diffusion models (Ho et al., 2020; Kingma et al., 2021) to develop an alternative VI scheme for MMF.

### 3.1. Generative Model

The generative model we consider is a modification of that in Section 2.1. Denote

$$\boldsymbol{x}_0 = \boldsymbol{y}, \quad \boldsymbol{x}_T = \boldsymbol{z}.$$

The generation of the data point $\boldsymbol{x}_0$ from the latent variable $\boldsymbol{x}_T$ follows a Markov process

$$\boldsymbol{x}_{t-1} = f_{t,\boldsymbol{\theta}}(\boldsymbol{x}_t) + \boldsymbol{v}_t, \quad t = T, T-1, \ldots, 1,$$

where $\boldsymbol{x}_t \in \mathbb{R}^{d_t}$ is the latent variable at layer $t$;

$$f_{t,\boldsymbol{\theta}}(\boldsymbol{x}_t) = \begin{cases} \rho(\boldsymbol{A}_1\boldsymbol{x}_1), & t = 1 & (7\text{a}) \\ \boldsymbol{B}_t\boldsymbol{x}_t + \boldsymbol{C}_t\rho(\boldsymbol{A}_t\boldsymbol{x}_t), & 2 \leq t \leq T-1; & (7\text{b}) \\ \boldsymbol{A}_T\boldsymbol{x}_T, & t = T & (7\text{c}) \end{cases}$$

$\boldsymbol{v}_t \sim \mathcal{N}(\boldsymbol{0}, \boldsymbol{\Sigma}_t)$ represents the modeling error at layer $t-1$ for $t \geq 2$ and noise for $t = 1$. The base latent variable $\boldsymbol{x}_T$

is distributed according to a given latent prior $p(\boldsymbol{x}_T)$, such as (3) and (4). The model parameter $\boldsymbol{\theta}$ contains all the $\boldsymbol{A}_t$'s, $\boldsymbol{B}_t$'s, $\boldsymbol{C}_t$'s, and $\boldsymbol{\Sigma}_t$'s.

Some justification should be provided for the model of $f_{t,\boldsymbol{\theta}}$. Eq. (7b) has a more general structure than its counterpart in (2), which basically considers $f_{t,\boldsymbol{\theta}} = \rho(\boldsymbol{A}_t \boldsymbol{x}_t)$. The merit will be clear as we proceed to diffusion VI. Eq. (7c) has no activation function $\rho$. This is to facilitate our VI development which will be described later. Taking out $\rho$ does not pose an issue: If $\boldsymbol{A}_T = \boldsymbol{I}$ and $\boldsymbol{\Sigma}_T \simeq 0$, then $\boldsymbol{x}_{T-1} \simeq \boldsymbol{x}_T$ and $\boldsymbol{x}_{T-2} \simeq f_{T-1,\boldsymbol{\theta}}(\boldsymbol{x}_T) + \boldsymbol{v}_{T-1}$. As a key assumption, we assume that the dimension of $\boldsymbol{x}_t$ is gradually decreasing:

$$M = d_0 \geq d_1 \geq d_2 \geq \ldots \geq d_T = N.$$

### 3.2. Dimension-Reducing Diffusion Model

Now we describe our proposed diffusion variational process. Consider the following process as our chosen variational process:

$$\boldsymbol{x}_t = \sqrt{a_t} \boldsymbol{U}_t^\top \boldsymbol{x}_{t-1} + \sqrt{1-a_t} \boldsymbol{e}_t, \ t = 1, ..., T-1, \quad (8\text{a})$$
$$\boldsymbol{x}_T \sim q_{\boldsymbol{\gamma}}(\boldsymbol{x}_T | \boldsymbol{x}_{T-1}), \quad (8\text{b})$$

where $0 < a_t < 1$; $\boldsymbol{e}_t \sim \mathcal{N}(\boldsymbol{0}, \boldsymbol{I})$ and the $\boldsymbol{e}_t$'s are independent; $\boldsymbol{U}_t \in \mathbb{R}^{d_{t-1} \times d_t}$ is semi-orthogonal; $q_{\boldsymbol{\gamma}}(\boldsymbol{x}_T | \boldsymbol{x}_{T-1})$ is latent-prior-dependent. In SCA (cf. (3)), we may choose $q_{\boldsymbol{\gamma}}(\boldsymbol{x}_T | \boldsymbol{x}_{T-1})$ as a Dirichlet distribution

$$q_{\boldsymbol{\gamma}}(\boldsymbol{x}_T | \boldsymbol{x}_{T-1}) = \text{Dir}(\boldsymbol{x}_T; \boldsymbol{\alpha}_{\boldsymbol{\gamma}}(\boldsymbol{x}_{T-1})),$$

where $\text{Dir}(\boldsymbol{x}; \boldsymbol{\alpha})$ denotes a Dirichlet distribution with parameter $\boldsymbol{\alpha}$; $\boldsymbol{\alpha}_{\boldsymbol{\gamma}}$ is a neural network with parameter $\boldsymbol{\gamma}$. For the non-negative latent prior in (4), we may choose $q_{\boldsymbol{\gamma}}$ as a Beta distribution. For ICA, we may choose a Gaussian distribution in the same way as that of the VAE (cf. Section 2.2). The selection criteria of $q_{\boldsymbol{\gamma}}(\boldsymbol{x}_T | \boldsymbol{x}_{T-1})$ are that (i) the support of $q_{\boldsymbol{\gamma}}(\boldsymbol{x}_T | \boldsymbol{x}_{T-1})$ is the same as that of the latent prior $p(\boldsymbol{x}_T)$; and that (ii) it has analytical expressions with its mean, covariance, and entropy. Table 3 in Appendix A.5 shows some examples. The variational model parameter $\boldsymbol{\phi}$ contains all the $\boldsymbol{a}_t$'s, $\boldsymbol{U}_t$'s, and $\boldsymbol{\gamma}$.

It is important to note that if $d_0 = \cdots = d_T$ and $\boldsymbol{U}_t = \boldsymbol{I}$, (8a) is exactly the diffusion model in the context of generative models. To the best of our knowledge, the dimension-reducing diffusion model in (8a) has not been considered before. The objective is not only to gradually add noise to the data point $\boldsymbol{x}_0$—a key part with the previous diffusion models—but also to gradually reduce the dimension. The dimension reduction feature is particularly relevant to MMF or the low-dimensional representation of high-dimensional data.

### 3.3. Dimension-Reducing Diffusion VI

Let us examine the ELBO under the above model. We have:

$$p_{\boldsymbol{\theta}}(\boldsymbol{x}_{0:T}) = p_{\boldsymbol{\theta}}(\boldsymbol{x}_0 | \boldsymbol{x}_1) \cdots p_{\boldsymbol{\theta}}(\boldsymbol{x}_{T-1} | \boldsymbol{x}_T) p(\boldsymbol{x}_T), \quad (9\text{a})$$
$$q_{\boldsymbol{\phi}}(\boldsymbol{x}_{1:T} | \boldsymbol{x}_0) = q_{\boldsymbol{\phi}}(\boldsymbol{x}_T | \boldsymbol{x}_{T-1}) \cdots q_{\boldsymbol{\phi}}(\boldsymbol{x}_1 | \boldsymbol{x}_0), \quad (9\text{b})$$

where

$$p_{\boldsymbol{\theta}}(\boldsymbol{x}_{t-1} | \boldsymbol{x}_t) = \mathcal{N}(\boldsymbol{x}_{t-1}; f_{t,\boldsymbol{\theta}}(\boldsymbol{x}_t), \boldsymbol{\Sigma}_t), \quad (10)$$

$$q_{\boldsymbol{\phi}}(\boldsymbol{x}_t | \boldsymbol{x}_{t-1}) = \mathcal{N}(\boldsymbol{x}_t; \sqrt{a_t} \boldsymbol{U}_t^\top \boldsymbol{x}_{t-1}, (1-a_t) \boldsymbol{I}),$$

for $t \leq T-1$. When applying (9a) and (9b) to the ELBO

$$\widehat{\mathcal{L}}(\boldsymbol{\theta}, \boldsymbol{\phi}; \boldsymbol{x}_0) = \mathbb{E}_{q_{\boldsymbol{\phi}}(\boldsymbol{x}_{1:T} | \boldsymbol{x}_0)} \left[ \log \frac{p_{\boldsymbol{\theta}}(\boldsymbol{x}_{0:T})}{q_{\boldsymbol{\phi}}(\boldsymbol{x}_{1:T} | \boldsymbol{x}_0)} \right], \quad (11)$$

it is natural to match $p_{\boldsymbol{\theta}}(\boldsymbol{x}_{t-1} | \boldsymbol{x}_t)$ and $q_{\boldsymbol{\phi}}(\boldsymbol{x}_t | \boldsymbol{x}_{t-1})$ and then to derive a multilayer ELBO expression; this is exactly what hierarchical VAEs do; see, e.g., Section 2.3 in (Luo, 2022). But this is not what diffusion VI does. It considers this alternative expression of $q_{\boldsymbol{\phi}}(\boldsymbol{x}_{1:T} | \boldsymbol{x}_0)$:

$$q_{\boldsymbol{\phi}}(\boldsymbol{x}_{1:T} | \boldsymbol{x}_0) = q_{\boldsymbol{\phi}}(\boldsymbol{x}_T | \boldsymbol{x}_0) \prod_{t=2}^{T} q_{\boldsymbol{\phi}}(\boldsymbol{x}_{t-1} | \boldsymbol{x}_t, \boldsymbol{x}_0), \quad (12)$$

which is obtained by applying $q_{\boldsymbol{\phi}}(\boldsymbol{x}_t | \boldsymbol{x}_{t-1}) = q_{\boldsymbol{\phi}}(\boldsymbol{x}_t | \boldsymbol{x}_{t-1}, \boldsymbol{x}_0)$ and $q_{\boldsymbol{\phi}}(\boldsymbol{x}_t | \boldsymbol{x}_{t-1}, \boldsymbol{x}_0) q_{\boldsymbol{\phi}}(\boldsymbol{x}_{t-1} | \boldsymbol{x}_0) = q_{\boldsymbol{\phi}}(\boldsymbol{x}_{t-1} | \boldsymbol{x}_t, \boldsymbol{x}_0) q_{\boldsymbol{\phi}}(\boldsymbol{x}_t | \boldsymbol{x}_0)$ to (9b). With (12), we can express $\widehat{\mathcal{L}}$ as

$$\widehat{\mathcal{L}}(\boldsymbol{\theta}, \boldsymbol{\phi}; \boldsymbol{x}_0) = \sum_{t=1}^{T} \widehat{\mathcal{L}}_t(\boldsymbol{\theta}, \boldsymbol{\phi}; \boldsymbol{x}_0), \quad (13)$$

where

$$\widehat{\mathcal{L}}_1 = \mathbb{E}_{q_{\boldsymbol{\phi}}(\boldsymbol{x}_1 | \boldsymbol{x}_0)} \left[ \log p_{\boldsymbol{\theta}}(\boldsymbol{x}_0 | \boldsymbol{x}_1) \right], \quad (14)$$

$$\widehat{\mathcal{L}}_t = \mathbb{E}_{q_{\boldsymbol{\phi}}(\boldsymbol{x}_{t-1}, \boldsymbol{x}_t | \boldsymbol{x}_0)} \left[ \log \frac{p_{\boldsymbol{\theta}}(\boldsymbol{x}_{t-1} | \boldsymbol{x}_t)}{q_{\boldsymbol{\phi}}(\boldsymbol{x}_{t-1} | \boldsymbol{x}_t, \boldsymbol{x}_0)} \right] \quad (15)$$

for $2 \leq t \leq T-1$, and

$$\widehat{\mathcal{L}}_T = \mathbb{E}_{q_{\boldsymbol{\phi}}(\boldsymbol{x}_{T-1}, \boldsymbol{x}_T | \boldsymbol{x}_0)} \left[ \log \frac{p(\boldsymbol{x}_T) p_{\boldsymbol{\theta}}(\boldsymbol{x}_{T-1} | \boldsymbol{x}_T)}{q_{\boldsymbol{\phi}}(\boldsymbol{x}_{T-1}, \boldsymbol{x}_T | \boldsymbol{x}_0)} \right]. \quad (16)$$

In the following, we will deal with each $\widehat{\mathcal{L}}_t$.

#### 3.3.1. LAYER−1 TERM $\widehat{\mathcal{L}}_1$

First, consider (14). Denote $\|\boldsymbol{x}\|_{\boldsymbol{\Sigma}}^2 = \boldsymbol{x}^\top \boldsymbol{\Sigma}^{-1} \boldsymbol{x}$. It can be shown that

$$-\widehat{\mathcal{L}}_1 \propto \underbrace{\frac{1}{2} \left( \mathbb{E}_{q_{\boldsymbol{\phi}}(\boldsymbol{x}_1 | \boldsymbol{x}_0)} \left[ \|\boldsymbol{x}_0 - \rho(\boldsymbol{A}_1 \boldsymbol{x}_1)\|_{\boldsymbol{\Sigma}_1}^2 \right] + \log |\boldsymbol{\Sigma}_1| \right)}_{r_1(\boldsymbol{\theta}, \boldsymbol{\phi}; \boldsymbol{x}_0)}$$

$$(17)$$

and that $q_{\boldsymbol{\phi}}(\boldsymbol{x}_1 | \boldsymbol{x}_0) = \mathcal{N}(\boldsymbol{x}_1; \sqrt{a_1} \boldsymbol{U}_1^\top \boldsymbol{x}_0, (1-a_1) \boldsymbol{I})$. This term can be readily handled by stochastic approximation.

### 3.3.2. LAYER–$t$ TERM $\widehat{\mathcal{L}}_t$, $2 \leq t \leq T - 1$

Second, consider (15). As a key result in diffusion models, $q_{\boldsymbol{\phi}}(\boldsymbol{x}_{t-1}, \boldsymbol{x}_t | \boldsymbol{x}_0)$ has an analytical expression. It can be shown that, for $2 \leq t \leq T - 1$,

$$q(\boldsymbol{x}_{t-1} | \boldsymbol{x}_t, \boldsymbol{x}_0) = \mathcal{N}(\boldsymbol{x}_{t-1}; \boldsymbol{\mu}_{t,\boldsymbol{\phi}}(\boldsymbol{x}_t, \boldsymbol{x}_0), \boldsymbol{\Psi}_{t,\boldsymbol{\phi}}), \quad (18)$$

where

$$\begin{aligned} \boldsymbol{\mu}_{t,\boldsymbol{\phi}} =& \frac{\sqrt{a_t}(1 - \bar{a}_{t-1})}{1 - \bar{a}_t} \boldsymbol{U}_t \boldsymbol{x}_t + \sqrt{\bar{a}_{t-1}} \bar{\boldsymbol{U}}_{t-1}^\top \boldsymbol{x}_0 \\ &+ \frac{\sqrt{\bar{a}_{t-1}}(\bar{a}_t - a_t)}{1 - \bar{a}_t} \boldsymbol{U}_t \boldsymbol{U}_t^\top \bar{\boldsymbol{U}}_{t-1}^\top \boldsymbol{x}_0; \end{aligned} \quad (19)$$

$$\boldsymbol{\Psi}_{t,\boldsymbol{\phi}} = (1 - \bar{a}_{t-1}) \left( \boldsymbol{I} - \frac{a_t - \bar{a}_t}{1 - \bar{a}_t} \boldsymbol{U}_t \boldsymbol{U}_t^\top \right); \quad (20)$$

$\bar{\boldsymbol{U}}_t = \boldsymbol{U}_1 \boldsymbol{U}_2 ... \boldsymbol{U}_t$; $\bar{a}_t = a_1 a_2 ... a_t$. This result is shown by using the fact that $q(\boldsymbol{x}_{t-1}, \boldsymbol{x}_t | \boldsymbol{x}_0)$ is Gaussian (for $2 \leq t \leq T - 1$). The derivations of (18) are relegated to Appendix A.1. Eq. (18) leads to a simplified result for $\widehat{\mathcal{L}}_t$. From (15) we can write

$$-\widehat{\mathcal{L}}_t = \mathbb{E}_{q_{\boldsymbol{\phi}}(\boldsymbol{x}_t | \boldsymbol{x}_0)} \underbrace{[D_{\mathrm{KL}}(q_{\boldsymbol{\phi}}(\boldsymbol{x}_{t-1} | \boldsymbol{x}_t, \boldsymbol{x}_0) || p_{\boldsymbol{\theta}}(\boldsymbol{x}_{t-1} | \boldsymbol{x}_t))]}_{:= D_t(\boldsymbol{x}_t; \boldsymbol{x}_0)},$$
$$(21)$$

where $D_{\mathrm{KL}}(q||p) = \int q(\boldsymbol{x}) \log (q(\boldsymbol{x})/p(\boldsymbol{x})) \, \mathrm{d}\boldsymbol{x}$ is the Kullback–Leibler (KL) divergence of two distributions $p$ and $q$. Let $\mathbb{S}_+^d$ and $\mathbb{S}_{++}^d$ denote the sets of all symmetric positive semidefinite and positive definite matrices in $\mathbb{R}^{d \times d}$, respectively. Consider the following lemma.

**Lemma 3.1.** *Let $\boldsymbol{\mu}_1$, $\boldsymbol{\mu}_2 \in \mathbb{R}^d$, $\boldsymbol{\Sigma}_1$, $\boldsymbol{\Sigma}_2 \in \mathbb{S}_{++}^d$, $\boldsymbol{\Psi} \in \mathbb{S}_+^d$. Let $p(\boldsymbol{x}) = \mathcal{N}(\boldsymbol{x}; \boldsymbol{\mu}_1, \boldsymbol{\Sigma}_1)$ and $q(\boldsymbol{x}) = \mathcal{N}(\boldsymbol{x}; \boldsymbol{\mu}_2, \boldsymbol{\Sigma}_2)$. Consider*

$$f(\boldsymbol{\Sigma}_1) = D_{\mathrm{KL}}(q||p) + \mathrm{tr}(\boldsymbol{\Sigma}_1^{-1} \boldsymbol{\Psi}).$$

*It holds that*

$$\min_{\boldsymbol{\Sigma}_1 \in \mathbb{S}_{++}^d} f(\boldsymbol{\Sigma}_1) = \frac{1}{2} \log \left| \boldsymbol{I} + \boldsymbol{\Sigma}_2^{-1/2}(\boldsymbol{W} + \boldsymbol{\Psi})\boldsymbol{\Sigma}_2^{-1/2} \right|,$$

*where $\boldsymbol{W} = (\boldsymbol{\mu}_2 - \boldsymbol{\mu}_1)(\boldsymbol{\mu}_2 - \boldsymbol{\mu}_1)^\top$, and the solution to the above problem is $\boldsymbol{\Sigma}_1^\star = \boldsymbol{\Sigma}_2 + \boldsymbol{W} + \boldsymbol{\Psi}$.*

The proof is provided in Appendix A.2. By applying Lemma 3.1 to $D_t$, and noting that $p_{\boldsymbol{\theta}}(\boldsymbol{x}_{t-1} | \boldsymbol{x})$ in (10) and $q_{\boldsymbol{\phi}}(\boldsymbol{x}_{t-1} | \boldsymbol{x}_t, \boldsymbol{x}_0)$ in (18) are Gaussian, we obtain

$$\min_{\boldsymbol{\Sigma}_t \in \mathbb{S}_{++}^d} D_t = \frac{1}{2} \log \left( 1 + \| f_{t,\boldsymbol{\theta}}(\boldsymbol{x}_t) - \boldsymbol{\mu}_{t,\boldsymbol{\phi}}(\boldsymbol{x}_t, \boldsymbol{x}_0) \|_{\boldsymbol{\Psi}_{t,\boldsymbol{\phi}}}^2 \right);$$

we have used $|\boldsymbol{I} + \boldsymbol{AB}| = |\boldsymbol{I} + \boldsymbol{BA}|$ to get the above equation. To facilitate the VI, we apply an approximation

$$\min_{\boldsymbol{\Sigma}_t \in \mathbb{S}_{++}^d} D_t \leq \frac{1}{2} \| f_{t,\boldsymbol{\theta}}(\boldsymbol{x}_t) - \boldsymbol{\mu}_{t,\boldsymbol{\phi}}(\boldsymbol{x}_t, \boldsymbol{x}_0) \|_{\boldsymbol{\Psi}_{t,\boldsymbol{\phi}}}^2 := \tilde{r}_t,$$

which is due to $\log(x) \leq x - 1$ for $x > 0$; this approximation is good if $\tilde{r}_t$ is small.

The above derivations show that VI intends to match $f_{t,\boldsymbol{\theta}}(\boldsymbol{x}_t)$ and $\boldsymbol{\mu}_{t,\boldsymbol{\phi}}(\boldsymbol{x}_t, \boldsymbol{x}_0)$. This motivates us to fix $\boldsymbol{B}_t$ and $\boldsymbol{C}_t$ in (7b) such that the structure of $f_{t,\boldsymbol{\theta}}(\boldsymbol{x}_t)$ matches that of $\boldsymbol{\mu}_{t,\boldsymbol{\phi}}(\boldsymbol{x}_t, \boldsymbol{x}_0)$ in (19). Specifically,

$$\begin{aligned} \boldsymbol{B}_t &= \frac{\sqrt{a_t}(1 - \bar{a}_{t-1})}{1 - \bar{a}_t} \boldsymbol{U}_t, \\ \boldsymbol{C}_t &= \frac{\sqrt{\bar{a}_{t-1}}(\bar{a}_t - a_t)}{1 - \bar{a}_t} \boldsymbol{U}_t \boldsymbol{U}_t^\top + \sqrt{\bar{a}_{t-1}} \boldsymbol{I}. \end{aligned} \quad (22)$$

With this choice, $\tilde{r}_t$ is simplified to

$$\tilde{r}_t = \frac{1}{2} \left\| \boldsymbol{C}_t \left( \bar{\boldsymbol{U}}_{t-1}^\top \boldsymbol{x}_0 - \rho(\boldsymbol{A}_t \boldsymbol{x}_t) \right) \right\|_{\boldsymbol{\Psi}_{t,\boldsymbol{\phi}}}^2. \quad (23)$$

In fact, $\tilde{r}_t$ can be further simplified to

$$\begin{aligned} \tilde{r}_t =& \frac{\bar{a}_{t-1}}{2(1 - \bar{a}_{t-1})} \left\| \bar{\boldsymbol{U}}_{t-1}^\top \boldsymbol{x}_0 - \rho(\boldsymbol{A}_t \boldsymbol{x}_t) \right\|_2^2 \\ &+ \frac{\bar{a}_t}{2(\bar{a}_t - 1)} \left\| \boldsymbol{U}_t^\top \left( \bar{\boldsymbol{U}}_{t-1}^\top \boldsymbol{x}_0 - \rho(\boldsymbol{A}_t \boldsymbol{x}_t) \right) \right\|_2^2. \end{aligned} \quad (24)$$

We relegate the derivation to Appendix A.3. This gives the following result

$$\max_{\boldsymbol{B}_t, \boldsymbol{C}_t, \boldsymbol{\Sigma}_t} \widehat{\mathcal{L}}_t \geq - \underbrace{\mathbb{E}_{q_{\boldsymbol{\phi}}(\boldsymbol{x}_t | \boldsymbol{x}_0)}[\tilde{r}_t(\boldsymbol{x}_t; \boldsymbol{x}_0)]}_{:= r_t(\boldsymbol{\theta}, \boldsymbol{\phi}; \, \boldsymbol{x}_0)}. \quad (25)$$

Note that $q_{\boldsymbol{\phi}}(\boldsymbol{x}_t | \boldsymbol{x}_0) = \mathcal{N}(\boldsymbol{x}_t; \sqrt{\bar{a}_t} \bar{\boldsymbol{U}}_t^\top \boldsymbol{x}_0, (1 - \bar{a}_t)\boldsymbol{I})$; this can be derived from $q_{\boldsymbol{\phi}}(\boldsymbol{x}_t | \boldsymbol{x}_{t-1})$. The function $r_t$ can be readily handled by stochastic approximation.

### 3.3.3. SOME INSIGHT

Let us pause a moment and try to get some intuitive insight. Eq. (25) suggests that VI intends to approximate

$$\bar{\boldsymbol{U}}_{t-1}^\top \boldsymbol{x}_0 \approx \rho(\boldsymbol{A}_t \boldsymbol{x}_t).$$

In particular, the left-hand side is a dimension-reduced $\boldsymbol{x}_0$ ($\bar{\boldsymbol{U}}_{t-1}$ is semi-orthogonal) while the right-hand side is a nonlinear low-dimension representation. Take the case of $\rho$ being a ReLU function as an example. We may want the dimension-reduced data point to be non-negative and possess a latent lower-dimensional structure. And this gradually happens from layer 1 to layer $T$.

### 3.3.4. LAYER–$T$ TERM $\widehat{\mathcal{L}}_T$

Third, consider (16). If $q_{\boldsymbol{\gamma}}(\boldsymbol{x}_T | \boldsymbol{x}_{T-1})$ takes a Gaussian form, then (16) may be handled in a similar way as in Section 3.3.2. If not, more work needs to be done; particularly, the key result in (18) no longer applies. Our derivations are as follows. We can decompose (16) as

$$-\widehat{\mathcal{L}}_T \propto \tilde{r}_T(\boldsymbol{\theta}, \boldsymbol{\phi}; \boldsymbol{x}_0) + r_{T+1}(\boldsymbol{\theta}, \boldsymbol{\phi}; \boldsymbol{x}_0), \quad (26)$$

where

$$\tilde{r}_T = \mathbb{E}_{q_\phi(\boldsymbol{x}_{T-1}|\boldsymbol{x}_0)}[\check{r}_T(\boldsymbol{\theta}, \boldsymbol{\phi}, \boldsymbol{x}_{T-1}; \boldsymbol{x}_0) \\ - \log q_\phi(\boldsymbol{x}_{T-1}|\boldsymbol{x}_0)]; \quad (27)$$

$$\check{r}_T = \mathbb{E}_{q_\gamma(\boldsymbol{x}_T|\boldsymbol{x}_{T-1})}\left[\log p_{\boldsymbol{\theta}}(\boldsymbol{x}_{T-1}|\boldsymbol{x}_T)\right];$$

$$r_{T+1} = \mathbb{E}_{q_\phi(\boldsymbol{x}_{T-1}|\boldsymbol{x}_0)}\left[H(q_\gamma(\boldsymbol{x}_T|\boldsymbol{x}_{T-1}))\right]; \quad (28)$$

$H(p(\boldsymbol{x})) = \int p(\boldsymbol{x}) \log p(\boldsymbol{x}) \mathrm{d}\boldsymbol{x}$ denotes the negative entropy of $p(\boldsymbol{x})$. Also note that

$$q_\phi(\boldsymbol{x}_{T-1}|\boldsymbol{x}_0) = \mathcal{N}(\boldsymbol{x}_{T-1}; \sqrt{\bar{a}_{T-1}}\bar{\boldsymbol{U}}_{T-1}^\top \boldsymbol{x}_0, (1-\bar{a}_{T-1})\boldsymbol{I}).$$

It can be shown that

$$\check{r}_T = \log \mathcal{N}\left(\boldsymbol{x}_{T-1}; \boldsymbol{A}_T \boldsymbol{\mu}_{T,\gamma}(\boldsymbol{x}_{T-1}), \boldsymbol{\Sigma}_T\right) \\ - \frac{1}{2}\mathrm{tr}\left(\boldsymbol{\Sigma}_T^{-1} \boldsymbol{A}_T \boldsymbol{\Psi}_{T,\gamma}(\boldsymbol{x}_{T-1}) \boldsymbol{A}_T^\top\right), \quad (29)$$

$$\boldsymbol{\mu}_{T,\gamma}(\boldsymbol{x}_{T-1}) = \mathbb{E}_{q_\gamma(\boldsymbol{x}_T|\boldsymbol{x}_{T-1})}[\boldsymbol{x}_T], \quad (30)$$

$$\boldsymbol{\Psi}_{T,\gamma}(\boldsymbol{x}_{T-1}) = \mathrm{Cov}_{q_\gamma(\boldsymbol{x}_T|\boldsymbol{x}_{T-1})}(\boldsymbol{x}_T); \quad (31)$$

see Appendix A.4. Consider the following lemma.

**Lemma 3.2.** *Consider the same settings in Lemma 3.1, except that $\boldsymbol{\mu}_1$ is changed to $\boldsymbol{\mu}_1(\boldsymbol{x})$, which is a function of $\boldsymbol{x}$. Let $\boldsymbol{R} = \mathbb{E}_{q(\boldsymbol{x})}\left[(\boldsymbol{x} - \boldsymbol{\mu}_1(\boldsymbol{x}))(\boldsymbol{x} - \boldsymbol{\mu}_1(\boldsymbol{x}))^\top\right]$. Suppose $\boldsymbol{R} + \boldsymbol{\Psi}$ is positive definite. Then*

$$\min_{\boldsymbol{\Sigma}_1 \in \mathbb{S}_{++}^d} f(\boldsymbol{\Sigma}_1) = \frac{1}{2}\log|\boldsymbol{\Sigma}_2^{-1/2}(\boldsymbol{R} + \boldsymbol{\Psi})\boldsymbol{\Sigma}_2^{-1/2}|,$$

*and the solution to the above problem is $\boldsymbol{\Sigma}_1^\star = \boldsymbol{R} + \boldsymbol{\Psi}$.*

The proof is provided in Appendix A.2. By applying Lemma 3.2 to (27) and (29), we obtain

$$\min_{\boldsymbol{\Sigma}_T \in \mathbb{S}_{++}^{d_T}} \tilde{r}_T = \frac{1}{2}\log\left|\frac{1}{1 - \bar{a}_{T-1}}(\boldsymbol{R} + \boldsymbol{G})\right|, \quad (32)$$

where

$$\boldsymbol{R} = \mathbb{E}_{q(\boldsymbol{x}_{T-1}|\boldsymbol{x}_0)}[(\boldsymbol{x}_{T-1} - \boldsymbol{\mu}_{T,\gamma}(\boldsymbol{x}_{T-1})) \\ (\boldsymbol{x}_{T-1} - \boldsymbol{\mu}_{T,\gamma}(\boldsymbol{x}_{T-1}))^\top],$$

$$\boldsymbol{G} = \mathbb{E}_{q(\boldsymbol{x}_{T-1}|\boldsymbol{x}_0)}\left[\boldsymbol{A}_T \boldsymbol{\Psi}_{T,\gamma}(\boldsymbol{x}_{T-1}) \boldsymbol{A}_T^\top\right].$$

We assume that $\boldsymbol{R} + \boldsymbol{G}$ is positive definite, which is a fairly mild assumption. Eq.(32) looks complicated. To facilitate VI, we consider

$$\min_{\boldsymbol{\Sigma}_T \in \mathbb{S}_{++}^{d_T}} \tilde{r}_T \leq \frac{1}{2}\mathrm{tr}\left(\frac{1}{1 - \bar{a}_{T-1}}(\boldsymbol{R} + \boldsymbol{G})\right) - \frac{d_T}{2}$$

$$\propto \frac{1}{2(1 - \bar{a}_{T-1})}\left(\mathbb{E}_{q_\phi(\boldsymbol{x}_{T-1}|\boldsymbol{x}_0)}\left[\|\boldsymbol{x}_{T-1} - \boldsymbol{\mu}_{T,\gamma}(\boldsymbol{x}_{T-1})\|_2^2 \right.\right.$$

$$\left.\left. + \mathrm{tr}\left(\boldsymbol{A}_T \boldsymbol{\Psi}_{T,\gamma}(\boldsymbol{x}_{T-1}) \boldsymbol{A}_T^\top\right)\right]\right) := r_T(\boldsymbol{\theta}, \boldsymbol{\phi}; \boldsymbol{x}_0),$$

$$\quad (33)$$

where we have used $\log(|\boldsymbol{A}|) \leq \mathrm{tr}(\boldsymbol{A}) - d$ for $\boldsymbol{A} \in \mathbb{S}_{++}^d$. As described previously, $q_\gamma(\boldsymbol{x}_T|\boldsymbol{x}_{T-1})$ is chosen such that $\boldsymbol{\mu}_{T,\gamma}(\boldsymbol{x}_{T-1})$, $\boldsymbol{\Psi}_{T,\gamma}(\boldsymbol{x}_{T-1})$, and $H(q_\gamma(\boldsymbol{x}_T|\boldsymbol{x}_{T-1}))$ have analytical expressions. Table 3 in Appendix A.5 gives some examples. The terms $r_T$ and $r_{T+1}$ can hence be handled by stochastic approximation.

### 3.3.5. REMAINING ASPECTS

Let us assemble the components together. The VI problem is

$$\min_{\boldsymbol{\phi}, \boldsymbol{\theta}} \frac{1}{L}\sum_{n=1}^L \sum_{t=1}^{T+1} r_t(\boldsymbol{\theta}, \boldsymbol{\phi}; \boldsymbol{y}_n) + \lambda \sum_{t=1}^{T-1}\left\|\boldsymbol{U}_t^\top \boldsymbol{U}_t - \boldsymbol{I}\right\|_\mathrm{F}^2, \quad (34)$$

where the $r_t$'s are given in (17), (25), (33), and (28); a regularization term is added to enforce the semi-orthogonality of $\boldsymbol{U}_t$'s; $\lambda \geq 0$ is given. Also, $\boldsymbol{\theta}$ is modified as $\boldsymbol{\theta} = \{\boldsymbol{\Sigma}_1, \boldsymbol{A}_1, ..., \boldsymbol{A}_T\}$. The latent variable estimate in (6) is given by

$$\widehat{\boldsymbol{z}}_n = \mathbb{E}_{q_\phi(\boldsymbol{x}_T|\boldsymbol{y}_n)}[\boldsymbol{x}_T] = \mathbb{E}_{q_\phi(\boldsymbol{x}_{T-1}|\boldsymbol{y}_n)}[\boldsymbol{\mu}_{T,\gamma}(\boldsymbol{x}_{T-1})], \quad (35)$$

where $\boldsymbol{\mu}_{T,\gamma}(\boldsymbol{x}_{T-1})$ is given in (30) and is assumed to have an analytical expression; Monte Carlo sampling may be used to compute $\widehat{\boldsymbol{z}}_n$. Alternatively, we can consider

$$\widehat{\boldsymbol{z}}_n \approx \boldsymbol{\mu}_{T,\gamma}\left(\mathbb{E}_{q_\phi(\boldsymbol{x}_{T-1}|\boldsymbol{y}_n)}[\boldsymbol{x}_{T-1}]\right)$$

$$= \boldsymbol{\mu}_{T,\gamma}\left(\sqrt{\bar{a}_{T-1}}\bar{\boldsymbol{U}}_{T-1}^\top \boldsymbol{y}_n\right),$$

which does not require Monte Carlo sampling.

## 4. Numerical Results

*Table 1.* Hyperspectral images for experiments.

| DATASET | $L$ | $d_T$ | $d_0$ |
|---|---|---|---|
| SAMSON | $95 \times 95$ | 3 | 156 |
| JASPER | $100 \times 100$ | 4 | 198 |
| APEX | $111 \times 122$ | 4 | 285 |
| URBAN | $307 \times 307$ | 6 | 162 |

In this section, we test the proposed DRD-VI for MMF with the latent priors in (3) and (4). For the uniform simplex prior in (3), the variational distribution $q_\gamma(\boldsymbol{x}_T|\boldsymbol{x}_{T-1})$ is chosen as a Dirichlet distribution

$$q_\gamma(\boldsymbol{x}_T|\boldsymbol{x}_{T-1}) = \mathrm{Dir}(\boldsymbol{x}_T; \boldsymbol{\alpha}_\gamma(\boldsymbol{x}_{T-1})),$$

where

$$\boldsymbol{\alpha}_\gamma(\boldsymbol{x}_{T-1}) = \exp(\boldsymbol{W}\boldsymbol{x}_{T-1})$$

is a one-layer network, and with $\boldsymbol{\gamma} = \boldsymbol{W}$. For the non-negative bounded uniform prior in (4), the variational distribution is chosen as a Beta distribution

$$q_\gamma(\boldsymbol{x}_T|\boldsymbol{x}_{T-1}) = \mathcal{B}(\boldsymbol{x}_T; \boldsymbol{\alpha}_\gamma(\boldsymbol{x}_{T-1}), \boldsymbol{\beta}_\gamma(\boldsymbol{x}_{T-1})),$$

*Table 2.* MSE averaged over all EMs (the best MSE among 10 independent trails/standard deviation).

| DATASET | SISAL | PRISM | CNNAEU | MISICNET | VASCA | DRD-VI |
|---------|-------|-------|--------|----------|-------|--------|
| SAMSON | 0.555/0.00 | 0.646/0.13 | 0.453/0.08 | 0.461/0.00 | 0.401/0.14 | **0.328**/0.00 |
| JASPER | 0.516/0.00 | 0.452/0.15 | 0.667/0.08 | 0.518/0.00 | 0.634/0.05 | **0.305**/0.09 |
| APEX | 0.743/0.00 | 0.645/0.15 | 0.812/0.06 | **0.413**/0.00 | 0.633/0.05 | 0.609/0.02 |
| URBAN | 0.796/0.02 | 0.824/0.12 | 0.700/0.12 | 0.955/0.00 | 0.785/0.03 | **0.677**/0.04 |

where

$$\boldsymbol{\alpha}_{\boldsymbol{\gamma}} = \exp(\boldsymbol{W}_{\alpha}\boldsymbol{x}_{T-1}), \quad \boldsymbol{\beta}_{\boldsymbol{\gamma}} = \exp(\boldsymbol{W}_{\beta}\boldsymbol{x}_{T-1}), \quad (36)$$

are one-layer networks, with $\boldsymbol{\gamma} = (\boldsymbol{W}_{\alpha}, \boldsymbol{W}_{\beta})$. The activation function $\rho$ is set as the ReLU function. In the experiments, we constrain $\boldsymbol{\Sigma}_1 = \sigma^2 \boldsymbol{I}$. We adopt the Adam algorithm (Kingma & Ba, 2015) for optimization.

### 4.1. Abundance Estimation in Hyperspectral Images

We first apply the proposed DRD-VI, with the uniform simplex prior (3), to the problem of estimating material abundance in hyperspectral images. This is a representative blind inverse problem in geoscience and remote sensing.

We briefly provide the background. In hyperspectral imaging, each image pixel is a $d_0$-dimensional vector capturing the electromagnetic reflectances of materials across $d_0$ spectral bands, known as spectral signatures or endmembers (EMs). Due to limited spatial resolution, a single pixel may contain mixed reflectances from multiple materials. The proportions of these EMs are modeled by a unit simplex variable $\boldsymbol{x}_T \in \mathbb{R}^{d_T}$ with $d_T \ll d_0$. Without the precise knowledge of the mixing process, abundance estimation aims to recover the abundance map $\boldsymbol{X}_T \in \mathbb{R}^{d_T \times L}$, where $L$ is the number of pixels. Here, each column of $\boldsymbol{X}_T$ represents the EMs' abundance in a single pixel, while each row shows the spatial distribution of an EM across the image. The task is to retrieve the low-dimensional simplex structures from the high-dimensional hyperspectral image.

We conduct experiments on four hyperspectral image datasets as listed in Table 1. We evaluate the mean squared error (MSE) defined as

$$\text{MSE}(\boldsymbol{X}_T, \boldsymbol{X}^{\star}) = \frac{1}{d_T}\sum_{i=1}^{d_T} \|\tilde{\boldsymbol{x}}_i - \tilde{\boldsymbol{x}}_i^{\star}\|_2/\|\tilde{\boldsymbol{x}}_i^{\star}\|_2$$

where $\boldsymbol{X}^{\star}$ is the reference ground truth provided by each dataset; $\tilde{\boldsymbol{x}}_i$ and $\tilde{\boldsymbol{x}}_i^{\star}$ are the $i$-th row of $\boldsymbol{X}_T$ and $\boldsymbol{X}^{\star}$, respectively. We use the Hungarian algorithm (Kuhn, 1955) to align the rows of $\boldsymbol{X}_T$ returned by algorithms with the rows of $\boldsymbol{X}^{\star}$.

The benchmark algorithms are as follows: Simplex identification via split augmented Lagrangian (SISAL) (Bioucas-Dias, 2009); PRobabIlistic SiMplex (PRISM) component

analysis method (Wu et al., 2021); and deep structures, CN-NAEU[1] (Palsson et al., 2020) and MiSiCNet[2] (Rasti et al., 2022). We also consider the VAE method with the log-norm variational distribution proposed in (Li et al., 2024), termed VASCA. VASCA employs a linear decoder. To make the comparison fair, we extend the linear decoder to the nonlinear generative model (2). The dimensions of the nonlinear decoder are the same as those of DRD-VI.

The experimental settings of DRD-VI, detailed in Appendix B.1, are consistent for all the tested hyperspectral images. Each algorithm is executed with 10 random initializations. Table 2 reports the overall MSE results, while the MSE contributions from each EM are provided in Appendix B.1.

Fig. 1 presents the estimated abundance map corresponding to the hyperspectral image Jasper. The abundance map results for other images are provided in Appendix B.1.

The results demonstrate that DRD-VI performs competitively, surpassing the state-of-the-art deep structures on some datasets and consistently outperforming VASCA.

### 4.2. Low-Dimensional Representation Learning

In this subsection, we consider DRD-VI with the nonnegative bounded uniform prior in (4). We compare DRD-VI with other state-of-the-art MMF methods following prior work on MMF (e.g., (Trigeorgis et al., 2016)) that evaluates MMF methods by analyzing the learned low-dimensional representations. Specifically, given a data matrix $\boldsymbol{X}_0$ with columns as i.i.d. samples, we apply clustering algorithms such as K-means to the low-dimensional representation matrix $\boldsymbol{X}_T$ produced by MF and MMF methods. The clustering results are evaluated using three standard metrics: adjusted rand index (ARI) (Hubert & Arabie, 1985), accuracy (Acc), and normalized mutual information (NMI) (Cai et al., 2005). Higher values of the three metrics indicate better performance, with a maximum of 1. It is believed that higher clustering performance indicates better-learned low-dimensional representations.

The benchmark algorithms are as follows: the one-layer semi-nonnegative matrix factorization (SNMF)[3](Ding et al.,

---

[1]Codes for CNNAEU.
[2]Codes for MiSiCNet.
[3]Codes for SNMF.

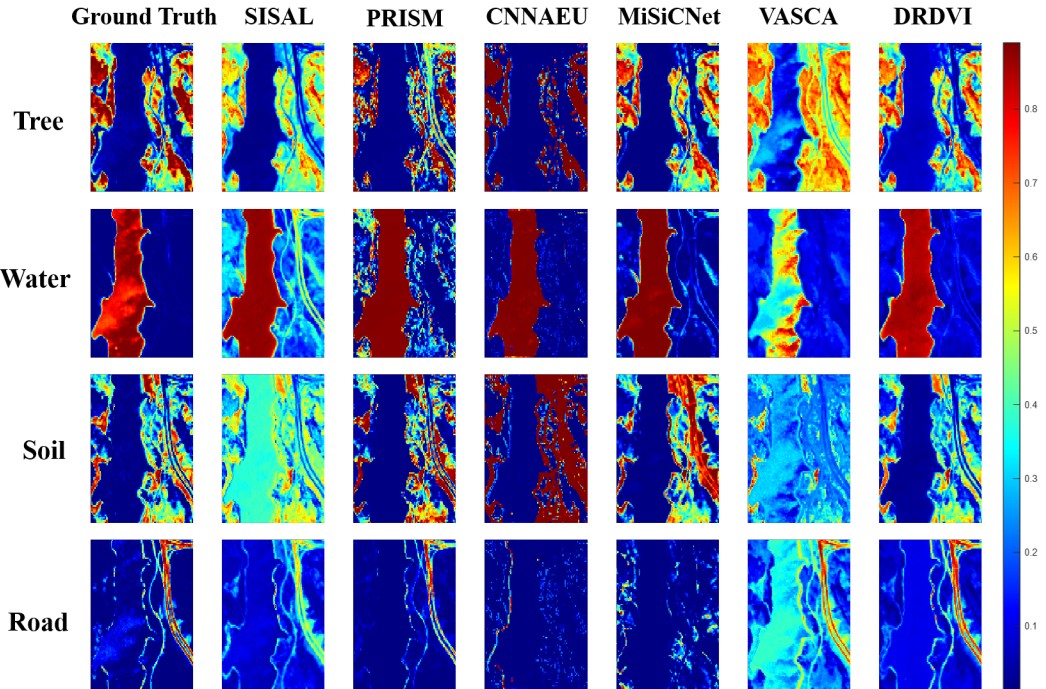

*Figure 1.* Estimated abundances for the hyperspectral image Jasper.

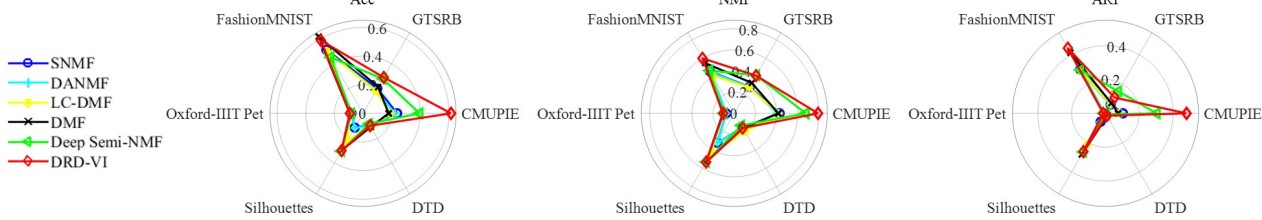

*Figure 2.* Performance comparison of the MF and MMF methods across the six datasets. The latent space dimensions equal 16 for gray image datasets and $16 \times 3$ for color ones.

2008), layer-centric deep matrix factorization (LC-DMF) (De Handschutter & Gillis, 2023), deep matrix factorization (DMF)[4] (Fan & Cheng, 2018), deep semi-nonnegative matrix factorization (Deep Semi-NMF)[5](Trigeorgis et al., 2016), and deep autoencoder-like nonnegative matrix factorization (DANMF)[6] (Ye et al., 2018). In the experiments, the model dimensions for the MMF methods and the dimension of the base latent variable for all methods are identical. We test the methods on six datasets: a freely available version

of CMU PIE (Sim et al., 2002), Caltech 101 Silhouettes[7], Fashion MNIST(Xiao, 2017), GTSRB(Houben et al., 2013), DTD(Cimpoi et al., 2014), and Oxford-IIIT Pet (Parkhi et al., 2012). Descriptions of the datasets and details of the experimental setups are provided in Appendix B.2.

The MF and MMF methods are applied to each dataset with 10 independent random initializations. For each trial, K-means clustering is performed on the learned representations with 50 independent random initializations, and the best clustering is recorded. The best results among the 10 trials are shown in Fig. 2. DRD-VI generally performs well and is

---

[4]Codes for DMF.

[5]Codes for Deep Semi-NMF.

[6]Codes for DANMF.

[7]Source of Caltech 101 Silhouettes.

comparable to, or in some cases outperforms, other state-of-the-art MMF methods. Due to space limitations, additional experimental results, including specific metric values and the effects of varying latent space dimensions, are provided in Appendix B.2.

## 5. Conclusion

This paper considered the application of diffusion model (DM)-based VI for MMF. We expanded on the idea of the existing variational DM, which assumes equal dimension with the latent variables, to propose a dimension-reducing variational DM for MMF. Each layer of the DM is associated with a layer of the MMF model, the latter of which can be seen as a shallow one-layer network (rather than a deep network in DMs for generative models). DMs are known to have the benefit of simple VI, and we turned that benefit to build a per-layer light-weight scheme for the VI of MMF. Experimental results showed that our proposed dimension-reducing DM-based VI scheme yields promising performance, suggesting the potential of variational DMs for MMF.

## Acknowledgements

This work was supported by a General Research Fund (GRF) of Hong Kong Research Grant Council (RGC) under Project ID CUHK 14203721.

## Impact Statement

This paper presents work whose goal is to advance the field of Machine Learning. There are many potential societal consequences of our work, none which we feel must be specifically highlighted here.

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

# A. Derivations and Proofs

## A.1. Derivation of (18)

Using Bayes' rule, we have

$$
\begin{aligned}
\log q(\boldsymbol{x}_{t-1}|\boldsymbol{x}_t, \boldsymbol{x}_0) &= \log \frac{q(\boldsymbol{x}_t|\boldsymbol{x}_{t-1}, \boldsymbol{x}_0)q(\boldsymbol{x}_{t-1}|\boldsymbol{x}_0)}{q(\boldsymbol{x}_t|\boldsymbol{x}_0)} \\
&= \log \frac{\mathcal{N}(\boldsymbol{x}_t; \sqrt{a_t}\boldsymbol{U}_t^\top \boldsymbol{x}_{t-1}, (1-a_t)\boldsymbol{I})\mathcal{N}\left(\boldsymbol{x}_{t-1}; \sqrt{\bar{a}_{t-1}}\bar{\boldsymbol{U}}_{t-1}^\top \boldsymbol{x}_0, (1-\bar{a}_{t-1})\boldsymbol{I}\right)}{\mathcal{N}\left(\boldsymbol{x}_t; \sqrt{\bar{a}_t}\bar{\boldsymbol{U}}_t^\top \boldsymbol{x}_0, (1-\bar{a}_t)\boldsymbol{I}\right)} \\
&= -\frac{1}{2}\left(\frac{a_t\boldsymbol{x}_{t-1}^\top \boldsymbol{U}_t\boldsymbol{U}_t^\top \boldsymbol{x}_{t-1} - 2\sqrt{a_t}\boldsymbol{x}_{t-1}^\top \boldsymbol{U}_t\boldsymbol{x}_t}{1-a_t} + \frac{\boldsymbol{x}_{t-1}^\top \boldsymbol{x}_{t-1} - 2\sqrt{\bar{a}_{t-1}}\boldsymbol{x}_{t-1}^\top \bar{\boldsymbol{U}}_{t-1}^\top \boldsymbol{x}_0}{1-\bar{a}_{t-1}}\right) + \text{constant} \\
&= -\frac{1}{2}\boldsymbol{x}_{t-1}^\top\left(\frac{a_t}{1-a_t}\boldsymbol{U}_t\boldsymbol{U}_t^\top + \frac{1}{1-\bar{a}_t}\boldsymbol{I}\right)\boldsymbol{x}_{t-1} + \boldsymbol{x}_{t-1}^\top\left(\frac{\sqrt{a_t}}{1-a_t}\boldsymbol{U}_t\boldsymbol{x}_t + \frac{\sqrt{\bar{a}_{t-1}}}{1-\bar{a}_{t-1}}\bar{\boldsymbol{U}}_{t-1}^\top \boldsymbol{x}_0\right) + \text{constant}.
\end{aligned}
\tag{37}
$$

We see that the above takes a quadratic form and it can be shown that $q(\boldsymbol{x}_{t-1}|\boldsymbol{x}_t, \boldsymbol{x}_0)$ is still a Gaussian distribution. The covariance matrix is given by

$$
\begin{aligned}
\boldsymbol{\Psi}_{t,\phi} &= \left(\frac{a_t}{1-a_t}\boldsymbol{U}_t\boldsymbol{U}_t^\top + \frac{1}{1-\bar{a}_{t-1}}\boldsymbol{I}\right)^{-1} \\
&= (1-\bar{a}_{t-1})\left(\boldsymbol{I} - \frac{a_t(1-\bar{a}_{t-1})}{1-a_t}\boldsymbol{U}_t\left(\boldsymbol{I} + \frac{a_t(1-\bar{a}_{t-1})}{1-a_t}\boldsymbol{U}_t^\top\boldsymbol{U}_t\right)^{-1}\boldsymbol{U}_t^\top\right) \\
&= (1-\bar{a}_{t-1})\left(\boldsymbol{I} - \frac{a_t(1-\bar{a}_{t-1})}{1-a_t}\left(1 + \frac{a_t(1-\bar{a}_{t-1})}{1-a_t}\right)^{-1}\boldsymbol{U}_t\boldsymbol{U}_t^\top\right) \\
&= (1-\bar{a}_{t-1})\left(\boldsymbol{I} - \frac{a_t - \bar{a}_t}{1-\bar{a}_t}\boldsymbol{U}_t\boldsymbol{U}_t^\top\right)
\end{aligned}
\tag{38}
$$

where we have used the matrix inverse formula

$$
(\boldsymbol{I} + \boldsymbol{X}\boldsymbol{Y})^{-1} = \boldsymbol{I} - \boldsymbol{X}(\boldsymbol{I} + \boldsymbol{Y}\boldsymbol{X})^{-1}\boldsymbol{Y}
\tag{39}
$$

for matrices $\boldsymbol{X}$ and $\boldsymbol{Y}$ with proper sizes in the second line, and the semi-orthogonality of $\boldsymbol{U}_t$ in the third line. The mean is given by

$$
\begin{aligned}
\boldsymbol{\mu}_{t,\phi}(\boldsymbol{x}_t, \boldsymbol{x}_0) &= \boldsymbol{\Psi}_{t,\phi}\frac{\sqrt{a_t}}{1-a_t}\boldsymbol{U}_t\boldsymbol{x}_t + \boldsymbol{\Psi}_{t,\phi}\frac{\sqrt{\bar{a}_{t-1}}}{1-\bar{a}_{t-1}}\bar{\boldsymbol{U}}_{t-1}^\top \boldsymbol{x}_0 \\
&= \left(\frac{\sqrt{a_t}(1-\bar{a}_{t-1})}{1-a_t} - \frac{a_t\sqrt{a_t}(1-\bar{a}_{t-1})^2}{(1-a_t)(1-\bar{a}_t)}\right)\boldsymbol{U}_t\boldsymbol{x}_t \\
&\quad + \sqrt{\bar{a}_{t-1}}\bar{\boldsymbol{U}}_{t-1}^\top \boldsymbol{x}_0 - \frac{\sqrt{\bar{a}_{t-1}}a_t(1-\bar{a}_{t-1})}{1-\bar{a}_t}\boldsymbol{U}_t\boldsymbol{U}_t^\top \bar{\boldsymbol{U}}_{t-1}^\top \boldsymbol{x}_0 \\
&= \frac{\sqrt{a_t}(1-\bar{a}_{t-1})}{1-\bar{a}_t}\boldsymbol{U}_t\boldsymbol{x}_t + \sqrt{\bar{a}_{t-1}}\bar{\boldsymbol{U}}_{t-1}^\top \boldsymbol{x}_0 + \frac{\sqrt{\bar{a}_{t-1}}(\bar{a}_t - a_t)}{1-\bar{a}_t}\boldsymbol{U}_t\boldsymbol{U}_t^\top \bar{\boldsymbol{U}}_{t-1}^\top \boldsymbol{x}_0.
\end{aligned}
\tag{40}
$$

## A.2. Proof of Lemma 3.1 and Lemma 3.2

First, we consider Lemma 3.1. It can be verified that

$$
\mathbb{E}_{q(\boldsymbol{x})}[\log p(\boldsymbol{x})] = -\frac{1}{2}\log|\boldsymbol{\Sigma}_1| - \frac{1}{2}\text{tr}(\boldsymbol{\Sigma}_1^{-1}\underbrace{\mathbb{E}_{q(\boldsymbol{x})}\left[(\boldsymbol{x}-\boldsymbol{\mu}_1)(\boldsymbol{x}-\boldsymbol{\mu}_1)^\top\right]}_{=\boldsymbol{R}}) - \frac{d}{2}\log(2\pi),
\tag{41}
$$

$$
\mathbb{E}_{q(\boldsymbol{x})}[\log q(\boldsymbol{x})] = -\frac{1}{2}\log|\boldsymbol{\Sigma}_2| - \frac{d}{2}(1 + \log(2\pi)).
\tag{42}
$$

The function $f$ can be written as

$$f(\boldsymbol{\Sigma}_1) = \frac{1}{2}\left(\log|\boldsymbol{\Sigma}_1| + \operatorname{tr}\left(\boldsymbol{\Sigma}_1^{-1}(\boldsymbol{R} + \boldsymbol{\Psi})\right)\right) + c \tag{43}$$

where $c = -\frac{1}{2}(\log|\boldsymbol{\Sigma}_2| + d)$. It is known that the solution to $\min_{\boldsymbol{\Sigma}_1 \in \mathbb{S}_{++}^d} f(\boldsymbol{\Sigma}_1)$ is uniquely given by $\boldsymbol{\Sigma}_1^\star = \boldsymbol{R} + \boldsymbol{\Psi}$ if $\boldsymbol{R} + \boldsymbol{\Psi}$ is positive definite (PD). To put this into context, consider the change of variable $\boldsymbol{Y} = \boldsymbol{\Sigma}_1^{-1}$. The corresponding objective function

$$f(\boldsymbol{Y}) = \frac{1}{2}\left(-\log|\boldsymbol{Y}| + \operatorname{tr}\left(\boldsymbol{Y}(\boldsymbol{R} + \boldsymbol{\Psi})\right)\right) + c$$

is convex, and its gradient equals

$$\nabla f(\boldsymbol{Y}) = \frac{1}{2}\left(-\boldsymbol{Y}^{-1} + \boldsymbol{R} + \boldsymbol{\Psi}\right);$$

see, e.g., (Boyd, 2004). It is easy to verify that

$$\boldsymbol{R} = \boldsymbol{\Sigma}_2 + \underbrace{(\boldsymbol{\mu}_2 - \boldsymbol{\mu}_1)(\boldsymbol{\mu}_2 - \boldsymbol{\mu}_1)^\top}_{=\boldsymbol{W}}. \tag{44}$$

Also, since $\boldsymbol{\Sigma}_2$ is PD, $\boldsymbol{R}$ is also PD. Putting the optimal solution $\boldsymbol{\Sigma}_1^\star$ into $f$ gives

$$f(\boldsymbol{\Sigma}^\star) = \frac{1}{2}\left(\log|\boldsymbol{R} + \boldsymbol{\Psi}| - \log|\boldsymbol{\Sigma}_2|\right) = \frac{1}{2}\log|\boldsymbol{\Sigma}_2^{-1/2}(\boldsymbol{R} + \boldsymbol{\Psi})\boldsymbol{\Sigma}_2^{-1/2}|, \tag{45}$$

and applying (44) to (45) gives the desired result.

Next, we consider Lemma 3.2. The proof is identical to the above, with the previous $\boldsymbol{R}$ being replaced by

$$\boldsymbol{R} = \mathbb{E}_{q(\boldsymbol{x})}\left[(\boldsymbol{x} - \boldsymbol{\mu}(\boldsymbol{x}))(\boldsymbol{x} - \boldsymbol{\mu}(\boldsymbol{x}))^\top\right].$$

### A.3. Derivation of (24)

Recall that the covariance matrices of $q_{\boldsymbol{\phi}}(\boldsymbol{x}_{t-1}|\boldsymbol{x}_t, \boldsymbol{x}_0)$ and $p_{\boldsymbol{\theta}}(\boldsymbol{x}_{t-1}|\boldsymbol{x}_t)$ are the same. The KL divergence can be written as

$$\begin{aligned}
&D_{\mathrm{KL}}\left(q_{\boldsymbol{\phi}}(\boldsymbol{x}_{t-1}|\boldsymbol{x}_t, \boldsymbol{x}_0)\|p_{\boldsymbol{\theta}}(\boldsymbol{x}_{t-1}|\boldsymbol{x}_t)\right) \\
=&\frac{1}{2}\left\|f_{t,\boldsymbol{\theta}}(\boldsymbol{x}_t) - \boldsymbol{\mu}_{t,\boldsymbol{\phi}}(\boldsymbol{x}_t, \boldsymbol{x}_0)\right\|_{\boldsymbol{\Psi}_{t,\boldsymbol{\phi}}}^2 + \text{constant} \\
=&\frac{1}{2}\left(f_{t,\boldsymbol{\theta}}(\boldsymbol{x}_t) - \boldsymbol{\mu}_{t,\boldsymbol{\phi}}(\boldsymbol{x}_t, \boldsymbol{x}_0)\right)^\top\left(\frac{a_t}{1-a_t}\boldsymbol{U}_t\boldsymbol{U}_t^\top + \frac{1}{1-\bar{a}_{t-1}}\boldsymbol{I}\right)\left(f_{t,\boldsymbol{\theta}}(\boldsymbol{x}_t) - \boldsymbol{\mu}_{t,\boldsymbol{\phi}}(\boldsymbol{x}_t, \boldsymbol{x}_0)\right) + \text{constant} \\
=&\frac{1}{2}\left(\frac{a_t}{1-a_t}\left\|\boldsymbol{U}_t^\top\left(f_{t,\boldsymbol{\theta}}(\boldsymbol{x}_t) - \boldsymbol{\mu}_{t,\boldsymbol{\phi}}(\boldsymbol{x}_t, \boldsymbol{x}_0)\right)\right\|_2^2 + \frac{1}{1-\bar{a}_{t-1}}\left\|f_{t,\boldsymbol{\theta}}(\boldsymbol{x}_t) - \boldsymbol{\mu}_{t,\boldsymbol{\phi}}(\boldsymbol{x}_t, \boldsymbol{x}_0)\right\|_2^2\right) + \text{constant.}
\end{aligned} \tag{46}$$

Based on (22), we can further write

$$\begin{aligned}
\frac{a_t}{1-a_t}\left\|\boldsymbol{U}_t^\top\left(f_{t,\boldsymbol{\theta}}(\boldsymbol{x}_t) - \boldsymbol{\mu}_{t,\boldsymbol{\phi}}(\boldsymbol{x}_t, \boldsymbol{x}_0)\right)\right\|_2^2 &= \frac{a_t\bar{a}_{t-1}}{1-a_t}\left\|\left(\boldsymbol{U}_t^\top - \frac{a_t(1-\bar{a}_{t-1})}{1-\bar{a}_t}\boldsymbol{U}_t^\top\right)\left(\bar{\boldsymbol{U}}_{t-1}^\top\boldsymbol{x}_0 - \rho(\boldsymbol{A}_t\boldsymbol{x})\right)\right\|_2^2 \\
&= \frac{\bar{a}_t(1-a_t)}{(1-\bar{a}_t)^2}\left\|\boldsymbol{U}_t^\top\left(\bar{\boldsymbol{U}}_{t-1}^\top\boldsymbol{x}_0 - \rho(\boldsymbol{A}_t\boldsymbol{x})\right)\right\|_2^2;
\end{aligned} \tag{47}$$

and

$$\begin{aligned}
&\frac{1}{1-\bar{a}_{t-1}}\|f_{t,\boldsymbol{\theta}}(\boldsymbol{x}_t) - \boldsymbol{\mu}_{t,\boldsymbol{\phi}}(\boldsymbol{x}_t, \boldsymbol{x}_0)\|_2^2 \\
=&\frac{\bar{a}_{t-1}}{1-\bar{a}_{t-1}}\left\|\left(\boldsymbol{I} - \frac{a_t(1-\bar{a}_{t-1})}{1-\bar{a}_t}\boldsymbol{U}_t\boldsymbol{U}_t^\top\right)\left(\bar{\boldsymbol{U}}_{t-1}^\top\boldsymbol{x}_0 - \rho(\boldsymbol{A}_t\boldsymbol{x})\right)\right\|_2^2 \\
=&\frac{\bar{a}_{t-1}}{1-\bar{a}_{t-1}}\left(\bar{\boldsymbol{U}}_{t-1}^\top\boldsymbol{x}_0 - \rho(\boldsymbol{A}_t\boldsymbol{x})\right)^\top\left(\boldsymbol{I} - \frac{(a_t-\bar{a}_t)(2-a_t-\bar{a}_t)}{(1-\bar{a}_t)^2}\boldsymbol{U}_t\boldsymbol{U}_t^\top\right)\left(\bar{\boldsymbol{U}}_{t-1}^\top\boldsymbol{x}_0 - \rho(\boldsymbol{A}_t\boldsymbol{x})\right) \\
=&\frac{\bar{a}_{t-1}}{1-\bar{a}_{t-1}}\left\|\bar{\boldsymbol{U}}_{t-1}^\top\boldsymbol{x}_0 - \rho(\boldsymbol{A}_t\boldsymbol{x})\right\|_2^2 - \frac{\bar{a}_t(2-a_t-\bar{a}_t)}{(1-\bar{a}_t)^2}\left\|\boldsymbol{U}_t^\top\left(\bar{\boldsymbol{U}}_{t-1}^\top\boldsymbol{x}_0 - \rho(\boldsymbol{A}_t\boldsymbol{x})\right)\right\|_2^2.
\end{aligned} \tag{48}$$

Adding (47) and (48) up gives

$$
D_{\mathrm{KL}}\left(q_{\boldsymbol{\phi}}(\boldsymbol{x}_{t-1}|\boldsymbol{x}_t,\boldsymbol{x}_0)\|p_{\boldsymbol{\theta}}(\boldsymbol{x}_{t-1}|\boldsymbol{x}_t)\right)
$$

$$
=\frac{\bar{a}_{t-1}}{2(1-\bar{a}_{t-1})}\left\|\bar{\boldsymbol{U}}_{t-1}^{\top}\boldsymbol{x}_0-\rho(\boldsymbol{A}_t\boldsymbol{x})\right\|_2^2+\frac{1}{2}\left(\frac{\bar{a}_t(1-a_t)}{(1-\bar{a}_t)^2}-\frac{\bar{a}_t(2-a_t-\bar{a}_t)}{(1-\bar{a}_t)^2}\right)\left\|\boldsymbol{U}_t^{\top}\left(\bar{\boldsymbol{U}}_{t-1}^{\top}\boldsymbol{x}_0-\rho(\boldsymbol{A}_t\boldsymbol{x})\right)\right\|_2^2 \qquad (49)
$$

$$
=\frac{\bar{a}_{t-1}}{2(1-\bar{a}_{t-1})}\left\|\bar{\boldsymbol{U}}_{t-1}^{\top}\boldsymbol{x}_0-\rho(\boldsymbol{A}_t\boldsymbol{x})\right\|_2^2+\frac{\bar{a}_t}{2(\bar{a}_t-1)}\left\|\boldsymbol{U}_t^{\top}\left(\bar{\boldsymbol{U}}_{t-1}^{\top}\boldsymbol{x}_0-\rho(\boldsymbol{A}_t\boldsymbol{x})\right)\right\|_2^2,
$$

which leads to the $\tilde{r}_t$ in (24).

### A.4. Derivation of (29)

We can write $\tilde{r}_T$ as

$$
\begin{aligned}
\check{r}_T ={}& \mathbb{E}_{q_{\boldsymbol{\gamma}}(\boldsymbol{x}_T|\boldsymbol{x}_{T-1})}\left[\log p_{\boldsymbol{\theta}}(\boldsymbol{x}_{T-1}|\boldsymbol{x}_T)\right] \\
={}& \mathbb{E}_{q_{\boldsymbol{\gamma}}(\boldsymbol{x}_T|\boldsymbol{x}_{T-1})}\left[\log\mathcal{N}(\boldsymbol{x}_{T-1};\boldsymbol{A}_T\boldsymbol{x}_T,\boldsymbol{\Sigma}_T)\right] \\
={}& \mathbb{E}_{q_{\boldsymbol{\gamma}}(\boldsymbol{x}_T|\boldsymbol{x}_{T-1})}\left[-\frac{1}{2}\|\boldsymbol{x}_{T-1}-\boldsymbol{A}_T\boldsymbol{x}_T\|_{\boldsymbol{\Sigma}_T}^2\right]-\frac{1}{2}\log\left(2\pi|\boldsymbol{\Sigma}_T|\right) \\
={}& \mathbb{E}_{q_{\boldsymbol{\gamma}}(\boldsymbol{x}_T|\boldsymbol{x}_{T-1})}\left[-\frac{1}{2}\left(\|\boldsymbol{x}_{T-1}\|_{\boldsymbol{\Sigma}_T}^2+\|\boldsymbol{A}_T\boldsymbol{x}_T\|_{\boldsymbol{\Sigma}_T}^2-2\boldsymbol{x}_{T-1}^{\top}\boldsymbol{\Sigma}_T^{-1}\boldsymbol{A}_T\boldsymbol{x}_T\right)\right]-\frac{1}{2}\log\left(2\pi|\boldsymbol{\Sigma}_T|\right) \\
={}& -\frac{1}{2}\left(\|\boldsymbol{x}_{T-1}\|_{\boldsymbol{\Sigma}_T}^2+\operatorname{tr}\left(\boldsymbol{A}^{\top}\boldsymbol{\Sigma}_T^{-1}\boldsymbol{A}_T\left(\boldsymbol{\mu}_{T,\boldsymbol{\gamma}}(\boldsymbol{x}_{T-1})\boldsymbol{\mu}_{T,\boldsymbol{\gamma}}(\boldsymbol{x}_{T-1})^{\top}+\boldsymbol{\Psi}_{T,\boldsymbol{\gamma}}(\boldsymbol{x}_{T-1})\right)\right)-2\boldsymbol{x}_{T-1}^{\top}\boldsymbol{\Sigma}_T^{-1}\boldsymbol{A}_T\boldsymbol{\mu}_{T,\boldsymbol{\gamma}}(\boldsymbol{x}_{T-1})\right) \\
& -\frac{1}{2}\log\left(2\pi|\boldsymbol{\Sigma}_T|\right) \\
={}& -\frac{1}{2}\left(\|\boldsymbol{x}_{T-1}\|_{\boldsymbol{\Sigma}_T}^2+\|\boldsymbol{A}_T\boldsymbol{\mu}_{T,\boldsymbol{\gamma}}(\boldsymbol{x}_{T-1})\|_{\boldsymbol{\Sigma}_T}^2-2\boldsymbol{x}_{T-1}^{\top}\boldsymbol{\Sigma}_T^{-1}\boldsymbol{A}_T\boldsymbol{x}_T\right)-\frac{1}{2}\log\left(2\pi|\boldsymbol{\Sigma}_T|\right) \\
& -\frac{1}{2}\operatorname{tr}\left(\boldsymbol{\Sigma}_T^{-1}\boldsymbol{A}_T\boldsymbol{\Psi}_{T,\boldsymbol{\gamma}}(\boldsymbol{x}_{T-1})\boldsymbol{A}^{\top}\right) \\
={}& \log\mathcal{N}(\boldsymbol{x}_{T-1};\boldsymbol{A}_T\boldsymbol{\mu}_{T,\boldsymbol{\gamma}}(\boldsymbol{x}_{T-1}),\boldsymbol{\Sigma}_T)-\frac{1}{2}\operatorname{tr}\left(\boldsymbol{\Sigma}_T^{-1}\boldsymbol{A}_T\boldsymbol{\Psi}_{T,\boldsymbol{\gamma}}(\boldsymbol{x}_{T-1})\boldsymbol{A}^{\top}\right),
\end{aligned}
$$

$$(50)$$

which gives the result in (29).

### A.5. Examples of applicable prior distributions

Table 3 presents examples of applicable distribution pairs. We clarify some notation. Given two vectors $\boldsymbol{x}$ and $\boldsymbol{y}$ of the same dimension, $\boldsymbol{x}\odot\boldsymbol{y}$ and $\boldsymbol{x}/\boldsymbol{y}$ denote the element-wise product and division, respectively. The symbols $\Gamma(\cdot)$ and $\psi(\cdot)$ denote the Gamma and Digamma functions, respectively. Given a vector $\boldsymbol{x}$, $|\boldsymbol{x}|$, $\Gamma(\boldsymbol{x})$, $\psi(\boldsymbol{x})$, $\log(\boldsymbol{x})$, $\exp(\boldsymbol{x})$ and $\boldsymbol{x}^2$ denote the element-wise operations of their scalar counterparts. The multivariate Beta distribution $\mathcal{B}(\boldsymbol{x};\boldsymbol{\alpha},\boldsymbol{\beta})$ is defined as $\prod_i\mathcal{B}(x_i;\alpha_i,\beta_i)$ where $\mathcal{B}(x;\alpha,\beta)$ is the probability density function of the one-dimensional Beta distribution; the same applies to the Laplace distribution. The symbol $c$ collects irrelevant constants.

## B. Experimental Setups and Additional Results

### B.1. Abundance Estimation in Hyperspectral Images

The settings of DRD-VI are listed in Table 4. For VASCA, the model dimensions, learning rate, batch size, and the number of epochs are set the same. Table 5 provides a more comprehensive MSE results for each hyperspectral image including contributions from each EM. Figs. 3-5 show the estimated abundance maps. The results show that the deep structure MiSiCNet performs well on the Apex dataset. DRD-VI consistently outperforms VASCA and is very competitive in general.

### B.2. Low-Dimensional Representation Learning

The datasets used are summarized in Table 6. Table 7 presents the experiment settings of the DRD-VI methods which are the same for all the datasets. The model dimensions of all other MMF methods are set the same as those of DRD-VI. Tables 8,

*Table 3.* Examples of distribution pairs.

| DISTRIBUTION | SETTING | EXPRESSION |
|---|---|---|
| LAPLACE (MEYER, 2021) | $p(\boldsymbol{x}_T)$ 
 $q_{\boldsymbol{\gamma}}(\boldsymbol{x}_T\|\boldsymbol{x}_{T-1})$ 
 $\boldsymbol{\mu}_{T,\boldsymbol{\gamma}}(\boldsymbol{x}_{T-1})$ 
 $\boldsymbol{\Psi}_{T,\boldsymbol{\gamma}}(\boldsymbol{x}_T)$ 
 $H(q_{\boldsymbol{\gamma}}(\boldsymbol{x}_T\|\boldsymbol{x}_{T-1}))$ | $\mathscr{L}(\boldsymbol{x}_T,\mathbf{1},\mathbf{1})$ 
 $\mathscr{L}(\boldsymbol{x}_T,\boldsymbol{\mu}_{\boldsymbol{\gamma}}(\boldsymbol{x}_{T-1}),\boldsymbol{b}_{\boldsymbol{\gamma}}(\boldsymbol{x}_{T-1}))$ 
 $\boldsymbol{\mu}_{\boldsymbol{\gamma}}$ 
 $\mathrm{DIAG}(2\boldsymbol{b}_{\boldsymbol{\gamma}}^2)$ 
 $\mathbf{1}^\top\left(\boldsymbol{b}\odot\exp\left(-\|\boldsymbol{\mu}_{\boldsymbol{\gamma}}\|/\boldsymbol{b}\right)+\|\boldsymbol{\mu}_{\boldsymbol{\gamma}}\|-\log\boldsymbol{b}\right)+c$ |
| DIRICHLET | $p(\boldsymbol{x}_T)$ 
 $q_{\boldsymbol{\gamma}}(\boldsymbol{x}_T\|\boldsymbol{x}_{T-1})$ 
 $\boldsymbol{\mu}_{T,\boldsymbol{\gamma}}(\boldsymbol{x}_{T-1})$ 
 $\boldsymbol{\Psi}_{T,\boldsymbol{\gamma}}(\boldsymbol{x}_T)$ 
 $H(q_{\boldsymbol{\gamma}}(\boldsymbol{x}_T\|\boldsymbol{x}_{T-1}))$ | $\mathrm{DIR}(\boldsymbol{x}_T,\mathbf{1})$ 
 $\mathrm{DIR}(\boldsymbol{x}_T,\boldsymbol{\alpha}_{\boldsymbol{\gamma}}(\boldsymbol{x}_{T-1}))$ 
 $\boldsymbol{\alpha}_{\boldsymbol{\gamma}}/(\mathbf{1}^\top\boldsymbol{\alpha}_{\boldsymbol{\gamma}})$ 
 $\left(\mathrm{DIAG}(\boldsymbol{\mu}_{T,\boldsymbol{\gamma}})-\boldsymbol{\mu}_{T,\boldsymbol{\gamma}}\boldsymbol{\mu}_{T,\boldsymbol{\gamma}}^\top\right)/\left(1+\mathbf{1}^\top\boldsymbol{\alpha}_{\boldsymbol{\gamma}}\right)$ 
 $(\boldsymbol{\alpha}_{\boldsymbol{\gamma}}-\mathbf{1})^\top\left(\psi(\boldsymbol{\alpha}_{\boldsymbol{\gamma}})-\psi\left(\mathbf{1}^\top\boldsymbol{\alpha}_{\boldsymbol{\gamma}}\right)\right)-\log\frac{\mathbf{1}^\top\Gamma(\boldsymbol{\alpha}_{\boldsymbol{\gamma}})}{\Gamma(\mathbf{1}^\top\boldsymbol{\alpha}_{\boldsymbol{\gamma}})}+c$ |
| BETA | $p(\boldsymbol{x}_T)$ 
 $q_{\boldsymbol{\gamma}}(\boldsymbol{x}_T\|\boldsymbol{x}_{T-1})$ 
 $\boldsymbol{\mu}_{T,\boldsymbol{\gamma}}(\boldsymbol{x}_{T-1})$ 
 $\boldsymbol{\Psi}_{T,\boldsymbol{\gamma}}(\boldsymbol{x}_T)$ 
 $H(q_{\boldsymbol{\gamma}}(\boldsymbol{x}_T\|\boldsymbol{x}_{T-1}))$ | $\mathcal{B}(\boldsymbol{x}_T;\mathbf{1},\mathbf{1})$ 
 $\mathcal{B}(\boldsymbol{x}_T;\boldsymbol{\alpha}_{\boldsymbol{\gamma}}(\boldsymbol{x}_{T-1}),\boldsymbol{\beta}_{\boldsymbol{\gamma}}(\boldsymbol{x}_{T-1}))$ 
 $\boldsymbol{\alpha}_{\boldsymbol{\gamma}}/(\boldsymbol{\alpha}_{\boldsymbol{\gamma}}+\boldsymbol{\beta}_{\boldsymbol{\gamma}})$ 
 $\mathrm{DIAG}\left((\boldsymbol{\alpha}_{\boldsymbol{\gamma}}\odot\boldsymbol{\beta}_{\boldsymbol{\gamma}})/\left(\boldsymbol{\alpha}_{\boldsymbol{\gamma}}+\boldsymbol{\beta}_{\boldsymbol{\gamma}}\right)^2/(\boldsymbol{\alpha}_{\boldsymbol{\gamma}}+\boldsymbol{\beta}_{\boldsymbol{\gamma}}+1)\right)$ 
 $(\boldsymbol{\alpha}_{\boldsymbol{\gamma}}-\mathbf{1})^\top\psi(\boldsymbol{\alpha}_{\boldsymbol{\gamma}})+(\boldsymbol{\beta}_{\boldsymbol{\gamma}}-\mathbf{1})^\top\psi(\boldsymbol{\beta}_{\boldsymbol{\gamma}})-(\boldsymbol{\alpha}_{\boldsymbol{\gamma}}+\boldsymbol{\beta}_{\boldsymbol{\gamma}}-2\mathbf{1})^\top\psi(\boldsymbol{\alpha}_{\boldsymbol{\gamma}}+\boldsymbol{\beta}_{\boldsymbol{\gamma}})-\mathbf{1}^\top\log\frac{\Gamma(\boldsymbol{\alpha}_{\boldsymbol{\gamma}})\odot\Gamma(\boldsymbol{\beta}_{\boldsymbol{\gamma}})}{\Gamma(\boldsymbol{\alpha}_{\boldsymbol{\gamma}}+\boldsymbol{\beta}_{\boldsymbol{\gamma}})}+c$ |

*Table 4.* Experimental settings of DRD-VI in abundance estimation.

| $[d_1, d_2, \ldots, d_T]$ | $\lambda$ | BATCH SIZE | EPOCH | LEARNING RATE |
|---|---|---|---|---|
| $[64, 32, 16, 8, d_T]$ | $10^5$ | $\mathrm{ROUND}(L/100)$ | 500 | 0.001 |

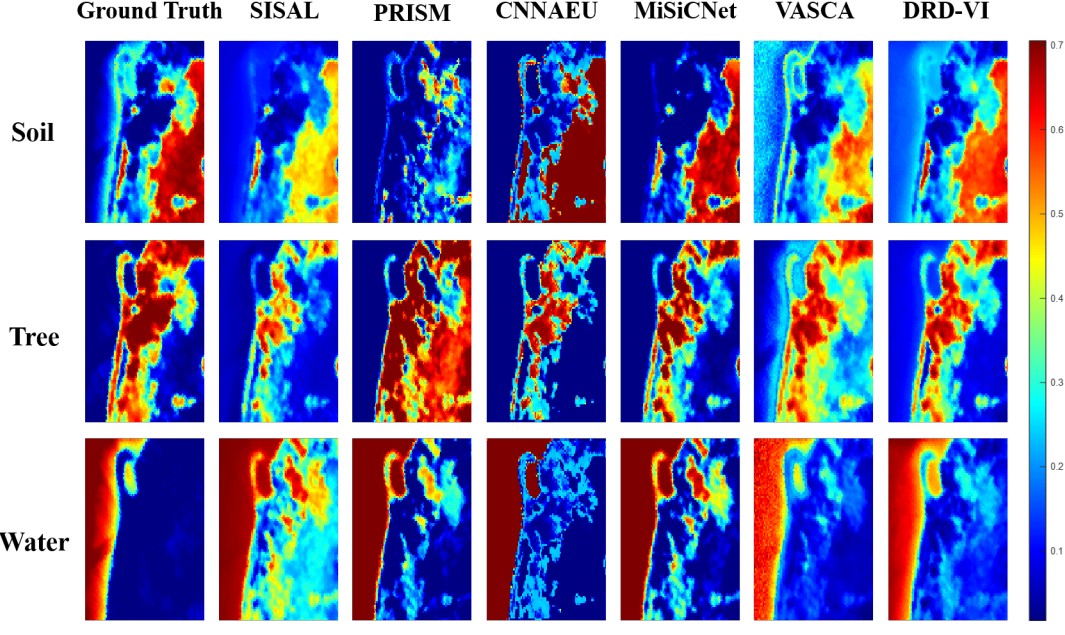

*Figure 3.* Estimated abundances for the hyperspectral image Samson.

9, and 10 present the results of applying the MF and MMF methods to the datasets, with the latent space dimension being 16 for gray image datasets and $16 \times 3$ for color ones. Note that the Deep Semi-NMF method uses a deterministic initialization.

We also present the results of varying the latent space dimension, with the other settings kept the same as before. Fig. 6,

*Table 5.* MSE results of abundance estimation (the best MSE among 10 independent trials/standard deviation).

| DATASET | ENDMEMBER | SISAL | PRISM | CNNAEU | MiSiCNet | VASCA | DRD-VI |
|---|---|---|---|---|---|---|---|
| SAMSON | SOIL | 0.387/0.00 | 0.426/0.21 | 0.410/0.11 | 0.351/0.00 | 0.408/0.25 | **0.269**/0.01 |
| | TREE | 0.494/0.00 | 0.630/0.22 | 0.469/0.07 | 0.335/0.00 | 0.380/0.09 | **0.314**/0.01 |
| | WATER | 0.785/0.00 | 0.882/0.12 | 0.480/0.14 | 0.697/0.00 | 0.416/0.10 | **0.401**/0.01 |
| | AVG. MSE | 0.555/0.00 | 0.646/0.13 | 0.453/0.08 | 0.461/0.00 | 0.401/0.14 | **0.328**/0.00 |
| JASPER | TREE | 0.432/0.00 | 0.498/0.14 | 0.450/0.13 | **0.190**/0.00 | 0.469/0.05 | 0.275/0.01 |
| | WATER | 0.457/0.00 | **0.142**/0.28 | 0.266/0.07 | 0.214/0.00 | 0.575/0.02 | 0.220/0.08 |
| | SOIL | 0.605/0.01 | 0.597/0.25 | 0.872/0.14 | 0.578/0.00 | 0.696/0.10 | **0.297**/0.05 |
| | ROAD | 0.569/0.00 | 0.570/0.15 | 1.079/0.16 | 1.093/0.00 | 0.794/0.22 | **0.428**/0.23 |
| | AVG. MSE | 0.516/0.00 | 0.452/0.15 | 0.667/0.08 | 0.518/0.00 | 0.634/0.05 | **0.305**/0.09 |
| APEX | ROAD | 0.911/0.00 | 1.173/0.29 | 1.596/0.29 | **0.648**/0.00 | 1.022/0.08 | 0.978/0.04 |
| | TREE | 0.543/0.00 | 0.488/0.13 | 0.567/0.09 | **0.255**/0.00 | 0.387/0.08 | 0.369/0.02 |
| | ROOF | 0.664/0.00 | 0.427/0.15 | 0.615/0.09 | **0.299**/0.00 | 0.619/0.05 | 0.389/0.02 |
| | WATER | 0.853/0.01 | 0.491/0.22 | 0.471/0.15 | **0.452**/0.00 | 0.503/0.09 | 0.698/0.01 |
| | AVG. MSE | 0.743/0.00 | 0.645/0.15 | 0.812/0.06 | **0.413**/0.00 | 0.633/0.05 | 0.609/0.02 |
| URBAN | ASPHALT | 0.816/0.02 | 0.923/0.17 | 0.626/0.10 | **0.539**/0.00 | 0.720/0.07 | 0.617/0.05 |
| | GRASS | 0.659/0.04 | 0.549/0.17 | 0.590/0.07 | 0.515/0.00 | 0.538/0.06 | **0.480**/0.04 |
| | TREE | 0.668/0.03 | **0.468**/0.21 | 0.496/0.07 | 0.587/0.00 | 0.695/0.08 | 0.542/0.01 |
| | ROOF | 1.004/0.11 | 1.024/0.45 | 0.581/0.09 | 0.638/0.00 | 0.895/0.09 | **0.566**/0.01 |
| | METAL | **0.907**/0.05 | 1.016/0.21 | 0.977/0.71 | 2.823/0.00 | 0.979/0.10 | 1.337/0.14 |
| | DIRT | 0.722/0.06 | 0.963/0.11 | 0.927/0.24 | 0.626/0.00 | 0.883/0.07 | **0.517**/0.04 |
| | AVG. MSE | 0.796/0.02 | 0.824/0.12 | 0.700/0.12 | 0.955/0.00 | 0.785/0.03 | **0.677**/0.04 |

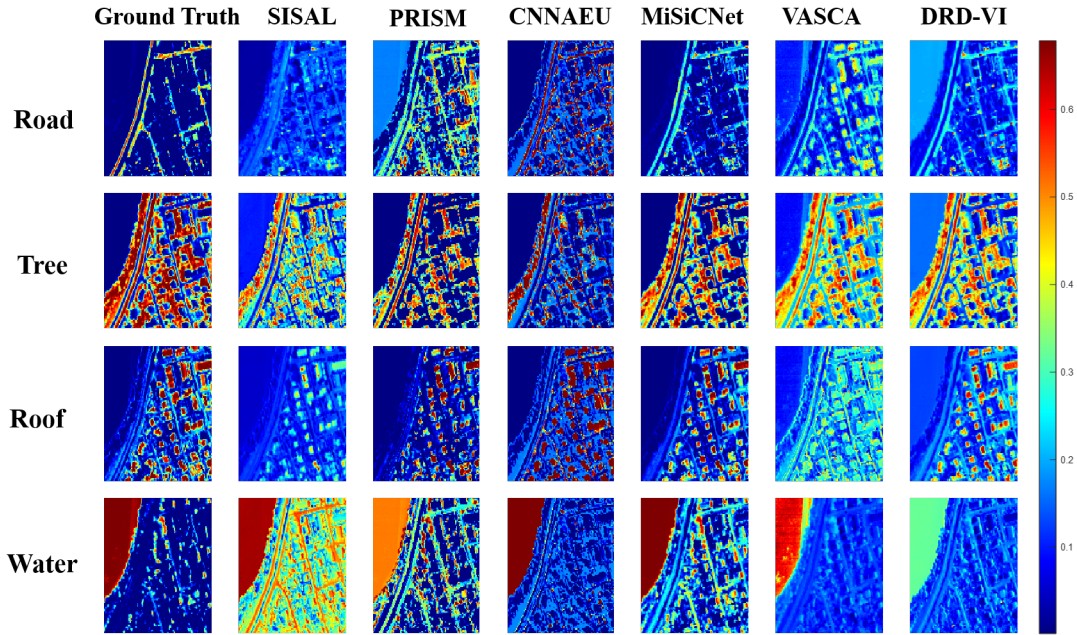

*Figure 4.* Estimated abundances for the hyperspectral image Apex.

7, and 8 present the results. DRD-VI is generally highly competitive compared to the other state-of-the-art methods and demonstrates a clearly better performance on the CMUPIE dataset.

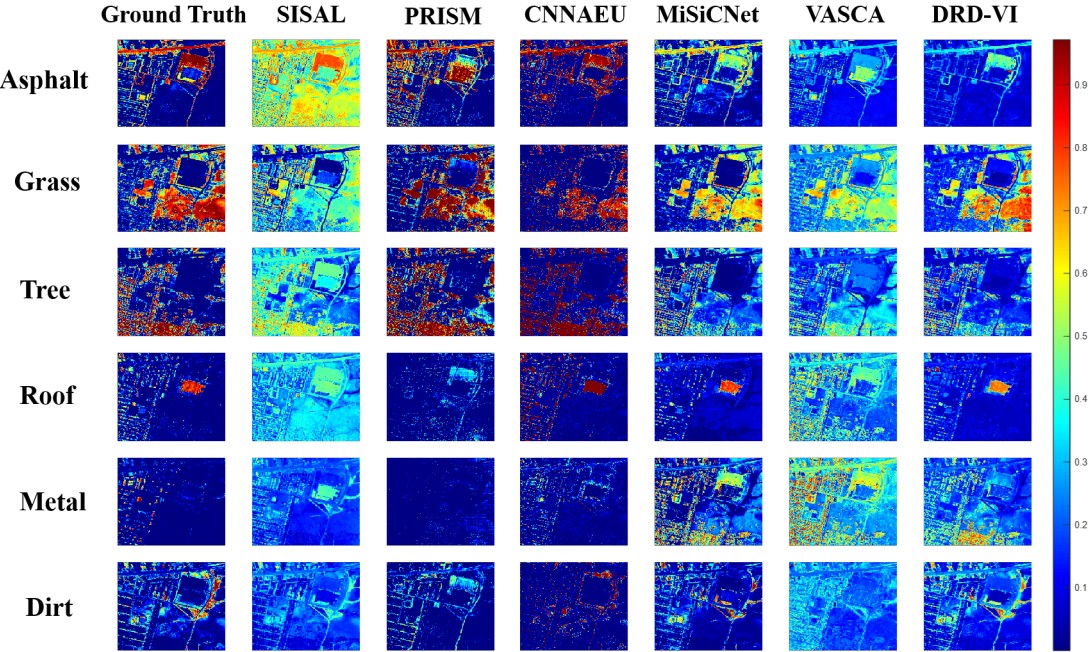

*Figure 5.* Estimated abundances for the hyperspectral image Urban.

*Table 6.* Image datasets used for low-dimensional representation learning.

| DATASET | DIMENSION | SAMPLES | CLUSTERS | DATA TYPE |
|---|---|---|---|---|
| CMU PIE (FREE VERSION) | $32 \times 32$ | 2856 | 68 | FACES |
| FASHION-MNIST (TESTING SET) | $28 \times 28$ | 10000 | 10 | CLOTHES |
| CALTECH 101 SILHOUETTES | $28 \times 28$ | 6407 | 101 | OBJECT SILHOUETTES |
| GTSRB (TESTING SET) | $32 \times 32 \times 3$ | 12630 | 43 | TRAFFIC SIGNS |
| OXFORD-IIIT PET (TESTING SET; RESIZED) | $40 \times 40 \times 3$ | 3680 | 37 | PETS |
| DTD (TESTING SET; RESIZED) | $50 \times 50 \times 3$ | 1880 | 47 | DESCRIBABLE TEXTURES |

*Table 7.* Low-dimensional representation learning experiment settings of DRD-VI.

| Data Type | $[d_1, d_2, \ldots, d_T]$ | $\lambda$ | Batch Size | Epoch | Learning Rate |
|---|---|---|---|---|---|
| Gray Image | $[256, 128, 64, 32, 16]$ | $10^6$ | ROUND($L/100$) | 500 | 0.001 |
| Color Image | $[256, 128, 64, 32, 16] \times 3$ | $10^6$ | | | |

*Table 8.* Accuracy (the best result among 10 independent trials/standard deviation).

| METHODS | CMU PIE | GTSRB | FASHION-MNIST | OXFORD-IIIT PET | SILHOUETTES | DTD |
|---|---|---|---|---|---|---|
| SNMF | 0.237/0.01 | 0.199/0.01 | 0.513/0.01 | 0.081/0.00 | 0.115/0.00 | 0.094/0.00 |
| DANMF | 0.223/0.01 | 0.186/0.00 | 0.497/0.00 | 0.079/0.00 | 0.118/0.00 | 0.098/0.00 |
| LC-DMF | 0.192/0.02 | 0.179/0.02 | 0.501/0.06 | 0.090/0.01 | 0.217/0.01 | **0.114**/0.00 |
| DMF | 0.179/0.01 | 0.211/0.00 | **0.620**/0.04 | 0.092/0.00 | 0.299/0.01 | 0.109/0.00 |
| DEEP SEMI-NMF | 0.395 | 0.279 | 0.452 | 0.081 | **0.304** | 0.085 |
| DRD-VI | **0.615**/0.01 | **0.290**/0.01 | 0.588/0.00 | **0.092**/0.00 | 0.300/0.01 | 0.100/0.00 |

*Table 9.* Normalized mutual information (the best result among 10 independent trials/standard deviation).

| METHODS | CMU PIE | GTSRB | FASHION-MNIST | OXFORD-IIIT PET | SILHOUETTES | DTD |
|---|---|---|---|---|---|---|
| SNMF | 0.428/0.01 | 0.288/0.01 | 0.457/0.01 | 0.076/0.00 | 0.320/0.00 | 0.157/0.00 |
| DANMF | 0.407/0.01 | 0.286/0.00 | 0.449/0.00 | 0.076/0.00 | 0.318/0.00 | 0.162/0.01 |
| LC-DMF | 0.421/0.03 | 0.281/0.03 | 0.414/0.06 | **0.109**/0.01 | 0.441/0.02 | **0.201**/0.00 |
| DMF | 0.417/0.02 | 0.332/0.01 | 0.549/0.02 | 0.108/0.00 | **0.535**/0.00 | 0.163/0.01 |
| DEEP SEMI-NMF | 0.677 | **0.416** | 0.467 | 0.100 | 0.530 | 0.132 |
| DRD-VI | **0.792**/0.01 | 0.412/0.01 | **0.601**/0.00 | 0.103/0.00 | 0.532/0.00 | 0.160/0.00 |

*Table 10.* Adjusted rand index (the best result among 10 independent trials/standard deviation).

| METHODS | CMU PIE | GTSRB | FASHION-MNIST | OXFORD-IIIT PET | SILHOUETTES | DTD |
|---|---|---|---|---|---|---|
| SNMF | 0.103/0.01 | 0.061/0.00 | 0.299/0.01 | 0.007/0.00 | 0.057/0.00 | 0.012/0.00 |
| DANMF | 0.092/0.00 | 0.058/0.00 | 0.272/0.00 | 0.007/0.00 | 0.059/0.00 | 0.014/0.00 |
| LC-DMF | 0.080/0.01 | 0.062/0.01 | 0.287/0.05 | 0.013/0.00 | 0.213/0.03 | 0.015/0.00 |
| DMF | 0.075/0.01 | 0.069/0.01 | 0.414/0.03 | 0.013/0.00 | **0.280**/0.01 | **0.020**/0.00 |
| DEEP SEMI-NMF | 0.304 | **0.151** | 0.296 | 0.011 | 0.263 | 0.008 |
| DRD-VI | **0.479**/0.01 | 0.108/0.01 | **0.445**/0.00 | **0.013**/0.00 | 0.261/0.01 | 0.016/0.00 |

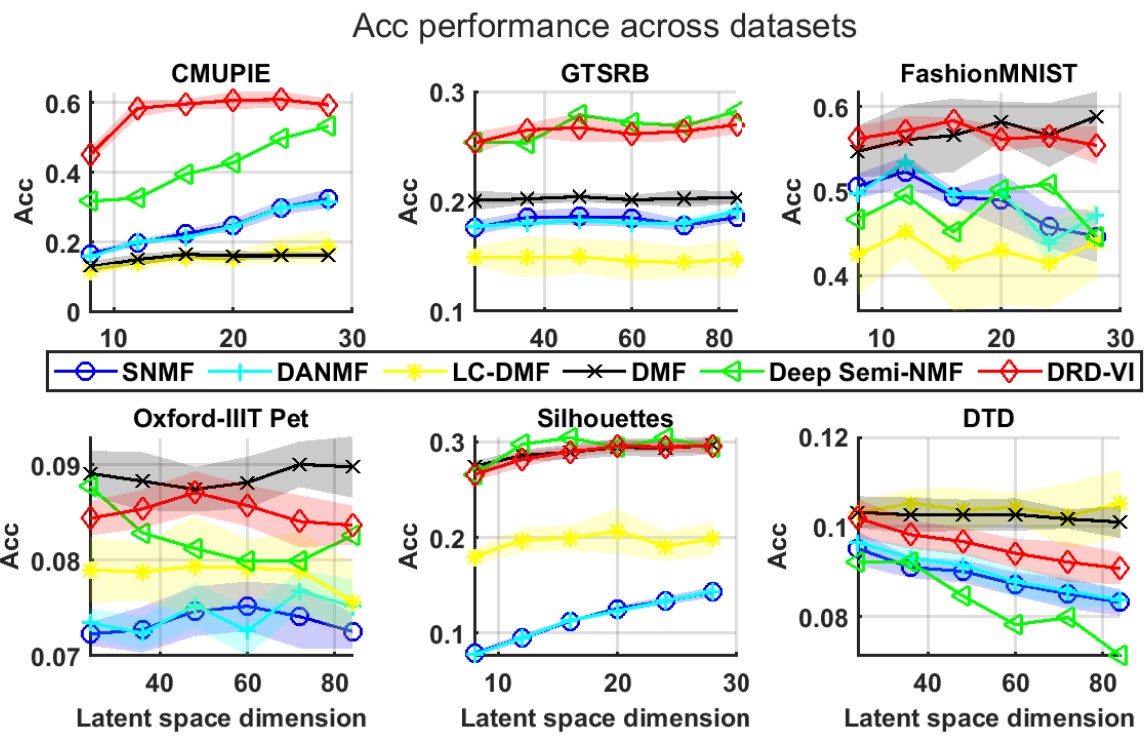

*Figure 6.* Accuracy vs. latent space dimensions.

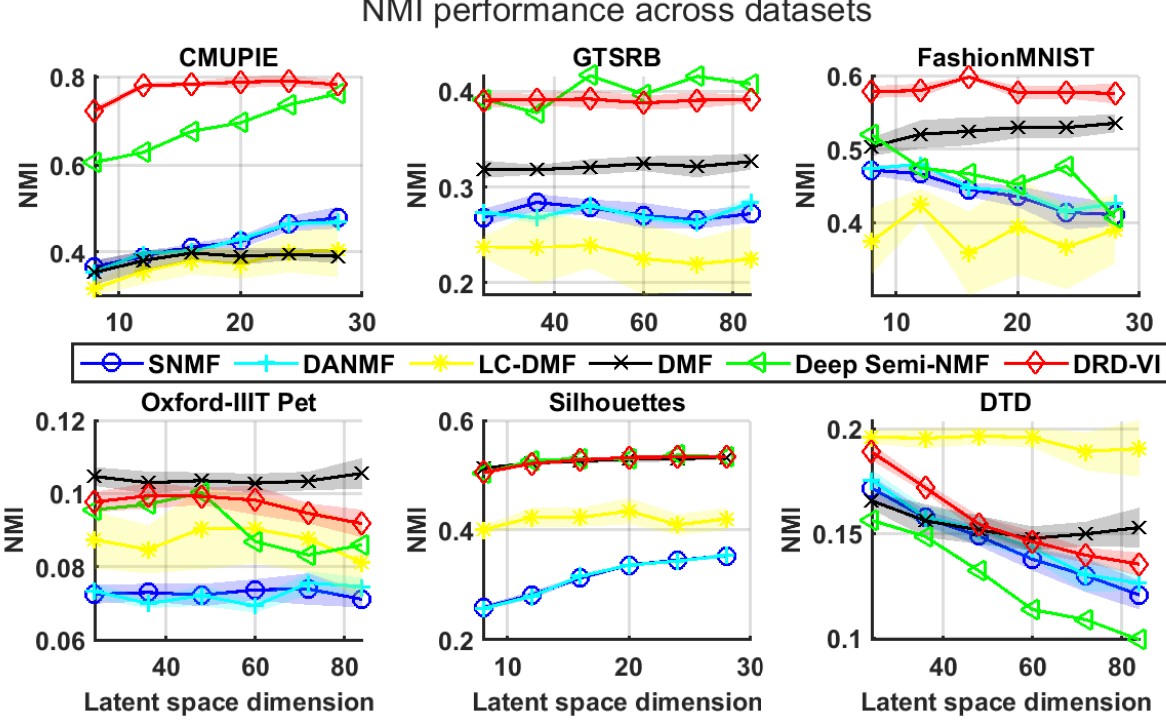

*Figure 7.* Normalized mutual information vs. latent space dimensions.

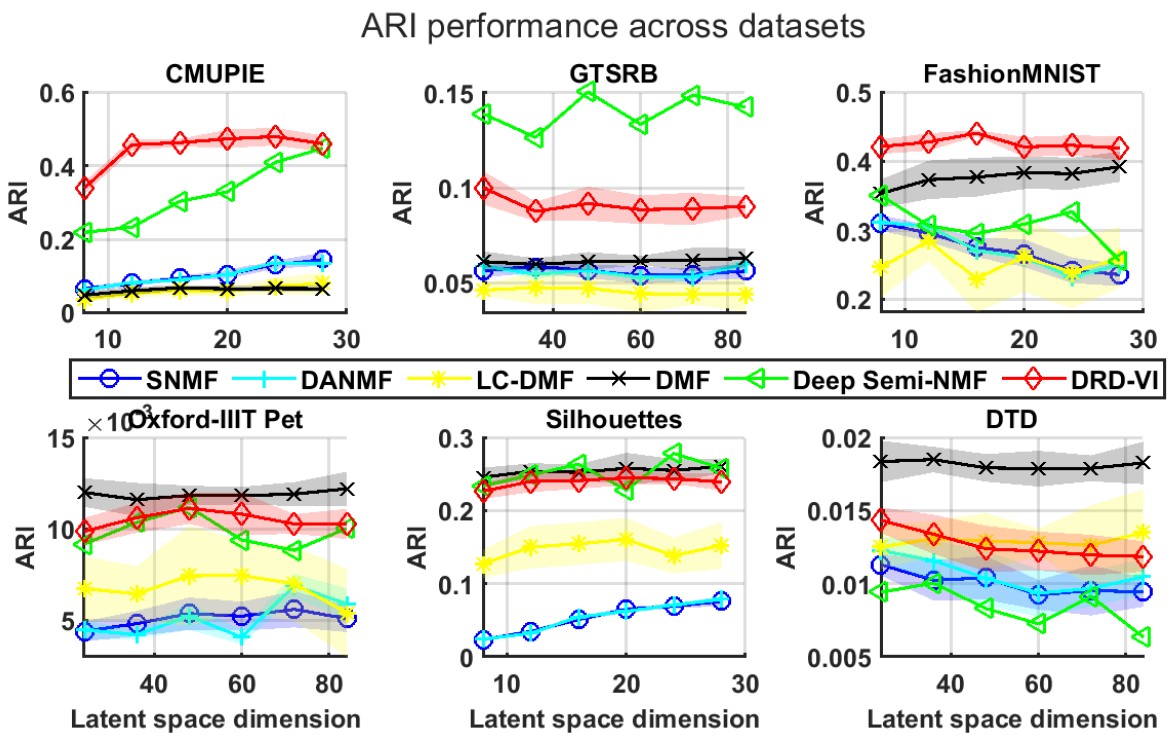

*Figure 8.* Adjusted rand index vs. latent space dimensions

