# OpenReview forum: "Multilayer Matrix Factorization via Dimension-Reducing Diffusion Variational Inference"
_ICML.cc/2025/Conference — ICML 2025 poster_

### Official Review · Reviewer_cESg · 2025-03-13

**Overall Recommendation:** 3

**Summary:**

This work presents a diffusion variational inference algorithm for multilayer matrix factorization. The authors treat each layer as a diffusion step. Another difference is that the dimension of the latent variable reduces with the layer depth, termed as dimension reduction diffusion VI. This is the nature of latent presentation learning in matrix factorization, and is different with diffusion VI with equal latent variable length. The dimension reduction is achieved by imposing an orthogonal transform matrix. Based on this, the authors derive the VI objective. Experiments are conducted on two tasks, abundance estimation of hyperspectral images and representation learning on several image datasets, in which the performance is further evaluated by clustering.

## Update after rebuttal

Since the authors address most concerns, I will keep the score.

**Claims And Evidence:**

The claims are supported by experiments.

**Essential References Not Discussed:**

No.

**Experimental Designs Or Analyses:**

The experimental design seems sound, but may be restricted to some simple datasets.

**Methods And Evaluation Criteria:**

Yes.

**Other Comments Or Suggestions:**

The overall organization is clear and easy to read. However, it contains numerous unnecessary and colloquial words. It could benefit from revising for conciseness. And there could be more space to present more results.

**Other Strengths And Weaknesses:**

**Strengths**

It introduces new inference techniques for deep matrix factorization. The model can be extended in many ways, for example changing the priors. The use of diffusion VI is promising and interesting to me. Therefore, I am learning to accept at this stage.

**Weakness**

1. From the perspective of inference techniques, the improvement may be not so significant. The main difference seems to be introducing the transform matrix $U$, in order to adjust the dimensions.

2. The experiments are conducted on simple datasets. I am not sure about the usefulness of the proposed models. Especially for the low dimensional representation learning, there are many simple yet powerful models. It is not sure whether the proposed model can be scaled to more complex and larger problems.

3. In the abundance estimation experiment, is the MSE evaluated on reconstruction error? Are there other metrics to evaluate the abundance estimation accuracy?

**Questions For Authors:**

1. To facilitate DRD-VI, the authors impose an orthogonal transform $U$ in the diffusion process. I am wondering if learning this $U$ along with VI parameters could make the optimization harder.

2. I am wondering what is the reason for adding the orthogonal constraint for $U$. And in practice, the orthogonality is introduced by regularization, which means that $U$ is not strictly orthogonal. Would this affect the results?

3. Does it require many layers for the diffusion VI?

**Relation To Broader Scientific Literature:**

The work mainly contributes to deep matrix factorization. This is a classical tool and can be related to many classical methods in signal processing.

**Theoretical Claims:**

There is no theoretical claim. There are many derivations about the final objective. I did not carefully check them.

---

> ### Author Rebuttal · Authors · 2025-04-01
>
> We are much obliged to you for your careful review and constructive comments. We will carefully take into account your comments in our revision, and we would like to discuss various aspects as follows.
>
> **Regarding “Other Strengths and Weaknesses”**
>
> Point 1: Thank you for expressing your view in the beginning that “the use of diffusion VI is promising and interesting to me.” In Point 1, you mentioned that the improvement from the inference viewpoint may not be so significant.
>
> Indeed, once you understand the principle of the equal-dimension diffusion models very well, it is perhaps not difficult to anticipate that dimensional-reducing diffusion models (more accurately, embedding dimensionality reduction directly into the diffusion model) could be constructed—at least intuitively. Still, the latter was not tried before. There are technical details to overcome, and some of them are not trivial. We would say that we start with a simple idea, but there are non-trivial details to work out, both in the development and in experiments.
>
> Point 2: As with some fundamental research, at this stage we focus on developing a concept (specifically, a new inference technique for MMF) and providing proof of concepts. As future work, the application of the concept to more complex datasets could be considered. Also, the hyperspectral unmixing problem we demonstrated in this paper is an important representative application in the context of hyperspectral remote sensing.
>
> Point 3: In addition to MSE, researchers in hyperspectral remote sensing also consider the abundance angle distance (AAD), which uses the angle of two vectors as the measure of similarity. We have provided AAD in the anonymous GitHub repository；please see Table [2](https://github.com/AnonymousPaper-Submission/AnonymousPaper-Submission-ICML2025-rebuttal) there.
>
>
> **Regarding “Other Comments Or Suggestions”**
>
> We agree and will try our best to better streamline the writing.
>
>
> **Regarding “Questions for Authors”**
>
> Point 1: Yes, optimization with semi-orthogonal matrix constraints is more difficult in principle. But that does not stop researchers from trying. In machine learning, for example,
>
> [1] Moustapha Cisse et al. "Parseval Networks: Improving Robustness to Adversarial Examples", ICML 2017\
> [2] Nitin Bansal et al. "Can We Gain More from Orthogonality
> Regularizations in Training Deep CNNs?", NeurIPS 2018,
>
> researchers have found success with their numerical results. The regularization method we use is essentially the same as that in the above references. Our empirical experience with the MMF application is that the regularization method works reasonably.
>
> Point 2: To answer your question, we provide some numerical results in Table [3 and 4](https://github.com/AnonymousPaper-Submission/AnonymousPaper-Submission-ICML2025-rebuttal). We observe that the $\boldsymbol{U}_t$'s are quite close to semi-orthogonality.
>
> Concerning the question why semi-orthogonal $\boldsymbol{U}_t$'s are used, one important reason is to simplify the variational process. It can be shown that if $\boldsymbol{U}_t$'s are not semi-orthogonal, the variational operations would be more complicated. In fact, if we do not control the rank of  $\boldsymbol{U}_t$ (which we do so via semi-orthogonality), there will be a lot of problems.
>
> Point 3: In the context of MMF, our experience is that we do not need as many layer as in generative models, which can be thousands. We would say tens for MMF. The concept introduced in this work does not pose a constraint on the number of layers one can use, however, and as future work it would be interesting to see other applications that require more layers.

---

> > ### Comment · Reviewer_cESg · 2025-04-07
> >
> > Thanks for the authors’ response. I admit the contributions and think the paper is interesting. I am overall positive with the paper and will consider it in the reviewer discussion phase.

---

> > > ### Author Response · Authors · 2025-04-09
> > >
> > > We sincerely thank Reviewer cESg for the constructive suggestions and the feedback on our response.

---

### Official Review · Reviewer_VFiD · 2025-03-16

**Overall Recommendation:** 4

**Summary:**

The paper introduces a novel diffusion-model based variational inference method for multilayer matrix factorization (MMF), using a dimension-reducing Markov chain as the noise. The method is evaluated on hyperspectral image unmixing, where it outperforms state-of-the-art MMF and deep learning methods in abundance estimation, and on low-dimensional representation learning, where it achieves competitive clustering performance.

**Claims And Evidence:**

Yes.

**Essential References Not Discussed:**

While the authors do not claim their paper is the first to propose dimension-reducing diffusion or diffusion for representation learning, it would be good to mention existing work on dimension-reducing diffusion such as Jing et al. 2022 and Zhang et al 2023.

Jing, Bowen, et al. "Subspace diffusion generative models." European Conference on Computer Vision. Cham: Springer Nature Switzerland, 2022.
Zhang, Han, et al. "Dimensionality-varying diffusion process." Proceedings of the IEEE/CVF conference on computer vision and pattern recognition. 2023.

**Experimental Designs Or Analyses:**

Yes. The experimental designs and analyses are sound.

**Methods And Evaluation Criteria:**

Yes.

**Other Comments Or Suggestions:**

- In the introduction section "in particular, for MMF, variational autoencoders (VAEs) appear to be the only viable solution" needs more clarification.
- The separation of the $\gamma$ and $\phi$ parameters are confusing. Since $\gamma$ is contained in $\phi$, maybe $q_\phi(x_T\mid x_{T-1})$ can be used instead of $q_\gamma(x_T\mid x_{T-1})$?
- There seems to be typos in eqations (15) and (16). It should be $q_\phi(x_{t-1}\mid x_t, x_0)$ and $q_\phi(x_{T-1}\mid x_{T}, x_0)$ instead of $q_\phi(x_{t-1}, x_t \mid x_0)$ and $q_\phi(x_{T-1}, x_{T}\mid x_0)$.
- The main paper contains too much derivation that does not directly contribute to the narrative of the paper. Many of the technical details can be moved to the appendix to make room for more discussions on the motivations and comparisons with existing methods. For example, section 3.3.3 can be expanded.

**Other Strengths And Weaknesses:**

- The paper presents the experiments very clearly. The setup on the abundance estimation problem is clear and easy to follow.
- The paper doesn't give a lot of insights and justifications for why the diffusion-based MMF is better or how it is different from hierarchical VAEs.

**Questions For Authors:**

1. In practice, what is the difference between the diffusion VI approach and the hierarchical VAE approach? What's the reason for preferring the diffusion approach over the VAE one?
2. What are some of the future directions of this work?

**Relation To Broader Scientific Literature:**

As far as I know, this paper is the first paper to leverage diffusion models for MMF. The proposed method achieves competitive or better performance than the previous state of the art methods on abundance estimation and representation learning.

**Theoretical Claims:**

I checked the formulation of the model and objective and they seem to be sound (apart from typos, see below). The details of the derivations are not thoroughly checked.

---

> ### Author Rebuttal · Authors · 2025-04-01
>
> We appreciate your time and effort in reviewing our work, and we are grateful for your generally positive feedback. We will take your advice to improve the paper. We also want to discuss some of the main points you raised.
>
> **Regarding “Essential References not Cited”**
>
> We agree, and thank you for providing us with the references. We will add a paragraph in the Introduction to describe existing works related to dimension-varying diffusion.
>
> We take this opportunity to give a reflection. The existing papers for dimension-varying diffusion are in the topic of generative models. They can be seen as concatenations of dimensionality-reduction processes and diffusion models. For the case of the references you provided, they concatenate multiple dimensionality-reduction processes and multiple diffusion models. On the other hand, we consider the topic of multilayer matrix factorization (MMF). We consider a single diffusion model and employ dimensionality reduction at each layer of the diffusion process, with the aim to develop a per-layer light-weight inference scheme. This makes the proposed method quite different from the dimension-varying diffusion methods we see in the prior generative modeling literature. However we agree that covering the prior literature would enhance the quality of this work, providing a broader coverage of related studies.
>
> **Regarding Comparison with Hierarchical VAEs**
>
> You mentioned that we didn’t give a lot of insights and justification on how diffusion-model based MMF differs from hierarchical VAEs (HVAEs). That is a good question, and we will try to improve the writing on this part in the revision. Initially, when we wrote this paper, we planned to cover the HVAEs in the main development (Section 3.3). However, we ended up abandoning it due to space limitation. It is in fact possible to apply HVAEs to the MMF model, associating each layer of the factorization model with one deep neural network for variational inference. We are not aware of any work on HVAEs for MMF, although it is possible. As described in [[1]](https://arxiv.org/pdf/2208.11970), the stochastic approximation in HVAEs may have larger variance when the number of layers is larger. This, together with the need to use one deep neural network for each layer of the factorization model, may not fit the purpose of per-layer light-weight operations well. One advantage of the diffusion model is that it uses a simple variational (diffusion) model to greatly simplify the variational process and sidestep the aforementioned high-variance issue [[1]](https://arxiv.org/pdf/2208.11970)—and this is our motivation for using diffusion models.
>
> It is also worthwhile to note the following: In principle, HVAEs can be directly applied to MMF. But the previous diffusion model cannot, because it assumes equal dimension with the latent variables. Our endeavor, simply speaking, is to make the MMF application possible.
>
> [1] Luo, Calvin. "Understanding diffusion models: A unified perspective." arXiv preprint arXiv:2208.11970 (2022).
>
> **Regarding “Other Comments or Suggestions”**
>
> We see your point and will try our best to streamline and make the writing more concise. The work consists of heavy details at some points, which can be better balanced, though we want to say that our writing style is to make the underlying assumptions and tricks clear to the reader; once again, we will try our best to balance. On “section 3.3.3 can be expanded,” you made a very good point and we agree. This, however, will be left as future work. We believe that from there we can develop more results.
>
> **Regarding “Questions to Authors”**
>
> Q1: We have run some simulations during this rebuttal period. Please see Table [1](https://github.com/AnonymousPaper-Submission/AnonymousPaper-Submission-ICML2025-rebuttal) in the anonymous GitHub repository. So far, our empirical finding is that, in general, the HVAE does not lead to performance improvement compared to the VAE. A possible reason is that the HVAE may require more careful training skills and more computations to achieve good performance.
>
> Q2: One future direction is to perform further analysis to expand or better understand the diffusion-based MMF interpretation in Section 3.3.3. We think that this part is quite unique to MMF in terms of providing interpretable explanations.

---

### Official Review · Reviewer_TTJj · 2025-03-25

**Overall Recommendation:** 4

**Summary:**

The paper presents presents a diffusion model (DM) based variational inference (VI) method for multilayer matrix factorization (MMF). They derive a variational process which is computationally efficient and lighter weight than other methods such as VAEs. Their method DRD-VI also reduces latent dimensionality at each step of the diffusion process, satisfying the requirement of matrix factorization to learn a lower dimensional representation of the data. The paper compares the performance of DRD-VI with multiple MF and MMF baselines on the problems of abundance estimation and low-dimensional representation learning with multiple datasets. They find DRD-VI outperforms the baselines on most datasets.

**Claims And Evidence:**

The claims are well supported and derivation of the proposed VI process is clear.

**Essential References Not Discussed:**

How do latent diffusion models fit in the context of this paper? The paper notes that diffusion models assume the latent space has the same dimensionality as the data, however there is existing work on latent diffusion models in a lower dimensional latent space including Rombach, Robin, et al. "High-resolution image synthesis with latent diffusion models." (Proceedings of the IEEE/CVF conference on computer vision and pattern recognition. 2022).

Another point in the paper is that DM-based VI has not been considered for MMF which may be the case, however there has been work done on related dimensionality reduction problems including diffusion based recommendation models such as Wang, Wenjie, et al. "Diffusion recommender model." (Proceedings of the 46th International ACM SIGIR Conference on Research and Development in Information Retrieval. 2023.) This paper in particular proposes a method L-DiffRec which compresses the data to a lower latent dimensionality and proceeds with the diffusion process in this latent space.

I believe both of these works as well as the literature on lower dimensional latent diffusion models as well as recommendation systems should be contextualized as related work in this paper.

**Experimental Designs Or Analyses:**

The experimental designs are valid as the authors evaluate on the established MMF blind inverse problem as well as the problem of low-dimensional representation learning using standard datasets such as CMU-PIE, Fashion-MNIST, etc.

**Methods And Evaluation Criteria:**

The proposed evaluation criteria are appropriate for the problem of MMF. The paper evaluates DRD-VI on the blind inverse problem of abundance estimation of hyperspectral images as well as low dimensional representation learning of six different datasets.

**Other Comments Or Suggestions:**

- Spelling error for section 2.2 title: "2.2. Varational Inference for MMF" should be "2.2. Variational Inference for MMF".
- In section 4.1, you misspell "MiSiCNet" as "MiSiNet" when introducing baselines (line 367).
- Table 2 in the paper has VAE listed as a method, should this be changed to VASCA as Figure 1 has it listed as VASCA?

**Other Strengths And Weaknesses:**

Strengths:
- The paper provides a thorough description and derivation of a VI method for diffusion model based MMF.
- The experimental results show DRD-VR is promising for multiple datasets.
- The method design choices are well justified.

Weaknesses:
- While the experimental results look good, the authors do not provide code which would help answer the questions regarding reproducibility and transparency.
- The authors should contextualize latent diffusion models and diffusion based recommendation models in their related work.

**Questions For Authors:**

- For the sake of reproducibility and future research in this area, can the authors release their code?
- The MiSiCNet results look better than DRD-VI for some settings (abundance estimation with the APEX dataset). Is there a reason or hypothesis for why this is the case?

**Relation To Broader Scientific Literature:**

The key contribution of the paper extends existing variational inference methods for diffusion models to the problem setting of MMF. By designing the VI process for diffusion to reduce dimensionality at each step with light weight layers, this paper connects to the broader areas of multilayer matrix factorization and diffusion model variational inference.

**Theoretical Claims:**

I did not thoroughly check each proof in the appendix, but the main equations in the paper look correct.

---

> ### Author Rebuttal · Authors · 2025-04-01
>
> We are very thankful for your thoughtful comments. We will do our best to revise and improve the paper. Here we would like to give a reflection on the main points you raised.
>
> **Regarding “Essential References not Cited”**
>
> Thank you for pointing out some references related to dimension-varying diffusion. We agree that we should cover that, and we plan to add a paragraph in the Introduction to describe the related works and how the current work differs.
>
> Let us take this opportunity to give an explanation. The related papers consider the context of generative models and generative recommendation models. They, from a high-level perspective, can be seen as concatenations of dimensionality-reduction processes and diffusion models. Dimensionality reduction is done outside of the diffusion process. Our proposed method considers the context of multilayer matrix factorization (MMF). We embed dimensionality reduction inside the diffusion process. Each layer of the factorization model (or each layer of the neural network) is associated with one layer of the diffusion process. These two features are not seen in the prior literature in generative models, to the best of our best knowledge, and are the unique parts of our development. In the context of MMF, the two features give us the opportunity to develop a per-layer light-weight scheme for MMF inference.
>
> While there are differences between the prior works and our current work, both fundamentally and in the topic of interest, we look back and agree that we should cover the prior papers in dimension-varying diffusion.
>
> **Regarding “Questions to Authors”**
>
> Q1: Absolutely, we can provide the codes. You can find the codes at the [anonymous GitHub repository](https://github.com/AnonymousPaper-Submission/ICML25-submission-codes-share).
>
> Q2: It could be difficult to give a reason when different methods use different models. MiSiCNet considers spatial correlations in its model, and this may give an edge to MiSiCNet in some instances.

---

> > ### Comment · Reviewer_TTJj · 2025-04-05
> >
> > Thank you for answering my questions and sharing your code.

---

> > > ### Author Response · Authors · 2025-04-09
> > >
> > > We sincerely thank Reviewer TTJj for the constructive review and the feedback on our response.

---

### Decision · Program_Chairs · 2025-05-01

**Decision:**

Accept (poster)

**Comment:**

This paper develops variational inference for multilayer matrix factorization using a dimension reducing diffusion model. The majority of the paper derives the variational lower bound using the diffusion approach of factorizing the posterior approximation in generative order rather than the markov reverse order.  The reviewers were all positive and the approach looks fresh.

Some other interesting references for this paper:

**Deep Exponential Families**: A multilayer matrix factorization with exponential families (https://arxiv.org/abs/1411.2581)
**Hierarchical Variational Models**: Improves variational bounds with auxiliary latent variables (https://arxiv.org/pdf/1511.02386)